# Factor-Wise Homogeneity of Slot-Attention for Continual Object-Centric Learning

**Ilmin Kang** [1]   **Hoyong Kim** [1]   **Seungju Bang** [1]   **Minwoo Kang** [1]   **Kangil Kim** [1]

## Abstract

While Object-Centric Learning has shown great promise in modular perception, its extension to Continual Learning remains underexplored. In this work, we observe that Slot Attention exhibits a distinctive behavior: it organizes latent representations into small and separated regions, each of which preserves identical factor states, crucially emerging not only in the current task but also across sequential tasks with novel factors. This *inter-task separation* offers significant advantages in continual learning, which typically suffers from severe object-wise forgetting. We refer to this phenomenon as *Factor-Wise Homogeneity*, and show that this intrinsic inter-task separation is crucial, serving as a key mechanism to prevent catastrophic forgetting in Continual Object-Centric Learning. However, despite its strong robustness, factor-wise homogeneity alone is insufficient due to the bottleneck in exploiting this separation at the decoder. To overcome this limitation and demonstrate the significance of our findings, we show that a minimal strategy *Decoder-only Post-Replay*, which freezes the factor-wise homogeneous representations and employs decoder-only fine-tuning, is sufficient. This work serves as a fundamental basis for understanding and leveraging the intrinsic dynamics of Slot Attention, offering essential insights for advancing object-centric systems.

## 1. Introduction

The ability to comprehend the compositional structure of complex scenes lies at the core of human intelligence. Humans naturally interpret visual input in terms of discrete, object-centric units, which facilitates continual generalization to novel and dynamic environments. Object-Centric Learning (OCL) (Greff et al., 2020) aims to replicate this capability by learning a set of disentangled representations, each corresponding to an individual object without the need for explicit supervision. OCL has recently gained increasing attention for its strong compositional generalization, with applications spanning object detection and segmentation (Lee et al., 2024; Zhang et al., 2021; Wang et al., 2023b; Zoran et al., 2021), image generation (Wang et al., 2023c; Akan & Yemez, 2025; Wu et al., 2023; Jiang et al., 2023), embodied AI (Hamdan & Güney, 2024; Baek et al.; Wu et al., b; Zadaianchuk et al.; Didolkar et al.), complex reasoning (Mondal et al.; 2024; Stammer et al.), and even text-based tasks (Park et al., 2024; Kim et al., 2023; Ma et al., 2023; Behjati & Henderson).

Despite these advances, existing OCL methods, such as Slot Attention (Locatello et al., 2020), typically operate under the assumption of a static, single-task environment. This limitation becomes critical where the model must acquire new object categories without suffering from catastrophic forgetting (McCloskey & Cohen, 1989). In this continuous setting, a fundamental challenge arises: *inter-task overlapping* (Buzzega et al., 2020a; Zhu et al., 2021; Ramasesh et al.; Kim et al., 2024b). As the model updates with new information, the latent representations of novel tasks tend to interfere with the regions occupied by previous tasks. This overlap between task-specific representations compromises the stability of model generalization across tasks by inducing destructive interference, where gradient updates minimizing the loss for the current task inevitably overwrite the latent features necessary for previous tasks, thereby accelerating catastrophic forgetting (Lopez-Paz & Ranzato, 2017a; Farajtabar et al., 2020; Ramasesh et al.).

Prior studies have demonstrated that Slot Attention mechanisms can identify features (Brady et al., 2023; Lachapelle et al., 2023; Kori et al., 2024) and uncover the compositional structure (Chang et al., 2022; Singh et al., b; Wiedemer et al.; Jung et al.) of individual objects, typically relying on assumptions or strong architectural inductive biases. However, understanding of whether or how Slot Attention inherently organizes its latent space remains limited. Fur-

---

[1]School of AI Convergence, Gwangju Institute of Science and Technology, Gwangju, South Korea. Correspondence to: Kangil Kim <kangil.kim.01@gmail.com>.

*Proceedings of the 43rd International Conference on Machine Learning*, Seoul, South Korea. PMLR 306, 2026. Copyright 2026 by the author(s).

thermore, while prior research has examined OCL generalization in out-of-distribution and zero-shot settings (Dittadi et al., 2021; Didolkar et al., 2025; Lachapelle et al., 2023; Wiedemer et al.; Chen et al., 2024) and the reusability of object-centric features (Pan et al.), the mechanism of structural collapse caused by task-wise overlapping under continuous updates has not yet been systematically investigated.

To address this gap, we investigate the representational dynamics of Slot Attention (Locatello et al., 2020) within the context of Continual Object-Centric Learning (COCL). Our analysis reveals a distinctive property: Slot Attention organizes its latent representations into small, well-separated regions, each consistently preserving the identical factor states. Crucially, this persists beyond single tasks and remains robust even across sequential training, offering significant advantages in continual learning where inter-task overlapping or confusion causes failures. We term this property *Factor-Wise Homogeneity*. and show that this intrinsic *inter-task separation* is crucial, serving as a key mechanism to prevent forgetting in COCL. However, despite its strong robustness, factor-wise homogeneity alone is insufficient due to the bottleneck in exploiting this separation at the decoder. To demonstrate the significance of our findings and overcome this limitation, we show that a minimal strategy *Decoder-only Post Replay* (DPR), which freezes the factor-wise homogeneous representations and employs decoder-only fine-tuning, is sufficient. We provide comprehensive validation on our COCL benchmarks and extend the evaluation to diverse environments. Furthermore, we investigate the Slot Attention architecture via Jacobian matrix and slot mode of gated RNNs, identifying the GRU as the key mechanism for these properties.

We summarize our contributions as follows: (1) We reveal the *factor-wise homogeneity* of Slot Attention: organizing latent representations into small, well-separated regions that consistently preserve factor states, maintaining robust *inter-task separation* under continual learning. (2) We demonstrate that this separation is a key mechanism to prevent catastrophic forgetting. By employing *Decoder-only Post-Replay* to address the bottleneck of the decoder, we prove that this intrinsic property is sufficient for COCL. (3) We introduce the first COCL benchmarks and conduct comprehensive analyses to validate our findings, extending our evaluation to more practical environments.

## 2. Related Work

**Object-Centric Learning and Slot Attention.** OCL aims to learn a set of disentangled representations, referred to as *slots*, where each slot corresponds to an individual object (Burgess et al., 2019; Greff et al., 2019; Locatello et al., 2020; Engelcke et al.), with Slot Attention (Locatello et al., 2020) being the most widely used framework. Previous

studies have primarily investigated the *identifiability* of slots through theoretical guarantees (Brady et al., 2023; Lachapelle et al., 2023; Kori et al., 2024) and architectural inductive biases (consistency) (Jia et al.; Didolkar et al.; Kori et al.; Liu et al., 2025). In parallel, Chang et al. (2022); Singh et al. (b); Jung et al.; Wu et al. (a); Mansouri et al. (2023); Wiedemer et al. have examined OCL in terms of factor-wise compositionality. Specifically, the Spatial Broadcast Decoder (SBD) (Watters et al., 2019) used in Slot Attention has been shown to identify latent variables and generalize to novel combinations of known factors via Cartesian-product extrapolation (Lachapelle et al., 2023; Wiedemer et al.). However, this capability fails on entirely unseen factors (Dittadi et al., 2021) and is limited by the decoder's capacity (Singh et al., a; Wu et al., 2023). Furthermore, while recent studies propose advanced variants to handle complex datasets (Singh et al., a; 2022; Seitzer et al.; Kakogeorgiou et al., 2024), they focus primarily on architectural modifications. Despite these extensive advances, a fundamental understanding of how Slot Attention inherently organizes its latent space remains limited.

**Object-Centric Learning for Continual Learning.** The core challenge of adapting Object-Centric Learning (OCL) to continual environments remains largely underexplored. Prior research has primarily focused on generalization in static or zero-shot settings, examining robustness to out-of-distribution shifts and zero-shot transferability (Dittadi et al., 2021; Didolkar et al., 2025). Theoretical works (Watters et al., 2019; Lachapelle et al., 2023; Wiedemer et al.) have established conditions for compositional generalization, ensuring extrapolation to novel compositions of known concepts. While some recent studies explore object-centric feature reusability for segmentation (Pan et al.) and topological homogeneity (Chen et al., 2024), the critical ability to continuously acquire novel object concepts without catastrophic forgetting remains an open challenge.

**Task-wise Separation in Continual Learning.** Achieving stable task-level separation is a significant challenge in continual learning, where representations of previous tasks are vulnerable to interference induced by novel task. Catastrophic forgetting driven from interference and overlapping representations between novel and previous tasks, results in biased unified classifier (Zhu et al., 2021). This instability is commonly attributed to logit or representation drift (Buzzega et al., 2020a; Caccia et al., 2022; Ramasesh et al.), semantic drift (Yu et al., 2020; Wu et al., 2025; Kim et al., 2024a), prototype shift coupled with boundary distortion (Zhu et al., 2021; Guo et al., 2023; Gomez-Villa et al., 2024), and class feature overlaps (Shmelkov et al., 2017; Kim et al., 2024b). Our findings reveal that Slot Attention exhibits *factor-wise homogeneity*, which facilitates task-wise separation. This intrinsic behavior mitigates the representational instability highlighted in prior research.

# 3. Continual Object-Centric Learning

## 3.1. Problem Formulation

**Object-Centric Learning and Slots.** Object-centric learning aims to decompose a scene into a set of distinct entities. To achieve this, slot-based approaches typically employ an auto-encoding framework comprising an encoder $f_\theta$ and a decoder $g_\phi$. This framework maps an image $x \in \mathcal{D}$ into a set of $K$ slots $\mathbf{z} = \{z_1, ..., z_K\}$ where $z_i \in \mathbb{R}^D$. The parameters $\theta$ and $\phi$ are optimized by minimizing the reconstruction error between the original image $x$ and the reconstruction $\hat{x}$ decoded from the slots.

**Inter-task Separation.** In the context of continual learning, we analyze the structure of the latent manifold. We first define the *Task-wise Cluster*, denoted as $\mathcal{R}_t$, for the $t$-th task $\mathcal{T}_t$. $\mathcal{R}_t$ represents the aggregate region formed by latent representations derived from $\mathcal{D}_t$:

$$\mathcal{R}_t \triangleq \{z \in \mathbb{R}^D \mid z \in f_\theta(x), \forall x \in \mathcal{D}_t\} \qquad (1)$$

Building upon this, we define *Inter-task Separation* as the distinct geometric arrangement of task-wise local clusters on the latent manifold. Specifically, this property implies a structural constraint: clusters corresponding to distinct tasks must avoid overlap and maintain *sufficient and clear margins* from one another. This geometric distinction guarantees that task-specific regions remain mutually exclusive, effectively preventing representation interference across tasks.

Conversely, any overlap between these task-wise clusters induces destructive interference, where gradient updates for the current task overwrite features essential for past ones, thereby accelerating catastrophic forgetting (Lopez-Paz & Ranzato, 2017a; Farajtabar et al., 2020; Ramasesh et al.).

**Definition of COCL.** We define Continual Object-Centric Learning (COCL) as learning from a sequential stream of tasks $\mathcal{T} = \{\mathcal{T}_t\}_{t=1}^M$. Each task $\mathcal{T}_t$ consists of a dataset $\mathcal{D}_t$ containing unlabeled multi-object images with objects from a specific class set $\mathcal{C}_t$, assuming mutual exclusivity across tasks (i.e., $\mathcal{C}_t \cap \mathcal{C}_k = \emptyset$ for all $t \neq k$). The primary objective is to perform object-centric learning while satisfying inter-task separation. This requires mapping representations exclusively to the current task-wise cluster $\mathcal{R}_t$ on latent manifold, ensuring no overlap with the latent spaces $\mathcal{R}_{k<t}$ of prior tasks to prevent catastrophic interference.

## 3.2. Datasets and evaluation frameworks

We introduce two benchmarks: (1) Continual-Tetrominoes, and (2) Continual-CLEVR. These datasets build upon the original Tetrominoes (Kabra et al., 2019) and CLEVR (Johnson et al., 2017) with additional augmentations to object (*shape*) classes. In this work, we focus on introducing novel

*shape* classes, as *shape* provides a broader range of variation compared to *position* or *color*, which are limited to bounded continuous ranges. We adopt three training and evaluation scenarios inspired by prior work (Shmelkov et al., 2017; Michieli & Zanuttigh, 2019; Cermelli et al., 2020), with modifications tailored for object-centric learning. The scenarios are defined as follows: (1) *Single Step addition of Two classes* (SST), (2) *Single Step addition of Multiple classes* (SSM), and (3) *Multi Step addition of Two classes* per step (MST). For evaluation, we follow (Dittadi et al., 2021) and mainly use Foreground Adjusted Rand Index (FG-ARI) (Rand, 1971), and provide additional validation using mean squared error (MSE) score Segmentation Covering (SC) (Arbelaez et al., 2010) and mean Segmentation Covering (mSC) (Engelcke et al.). Details of our datasets (including training scenarios) and evaluation metrics are demonstrated in Appendix C. and Appendix B.5.

# 4. Factor-Wise Homogeneity of Slot Attention

## 4.1. Preliminary

**Slot Attention** (Locatello et al., 2020) is composed of three components: (1) an encoder, (2) a slot attention module, and (3) a decoder. Given an input image $x_i \in \mathbb{R}^{C \times H \times W}$ of $t$-th task $\mathcal{T}_t$, the encoder $\mathbf{f}_t : \mathcal{X}_t \to \mathcal{Z}_t$ extracts $D$-dimensional features $z_i \in \mathbb{R}^{N \times D}$. Then, slot attention module $\mathbf{h}_t : \mathcal{Z}_t \to \mathcal{S}_t$ produces a set of $K$ vectors called *slots* $s_i \in \mathbb{R}^{K \times D_{slots}}$, where each slot corresponds to an individual object present in the input image $x_i$. Slots are initially sampled from the Gaussian distribution with learnable $\mu$ and $\sigma$. Each step performs dot-product attention (Luong et al., 2015) between $z_i$ and normalized slots from previous steps. Aggregated slots are updated using a Gated Recurrent Unit (GRU) (Chung et al., 2014) and a residual multi-layer perceptron (MLP) with normalization. Finally, each slot is independently decoded using a spatial broadcast decoder (Watters et al., 2019; Locatello et al., 2019) $\mathbf{g}_t : \mathcal{S}_t \to \hat{\mathcal{X}}_t$, and the model is trained to minimize mean squared error.

**Pseudo-labels for Empirical Analysis.** We investigate the distribution of slots trained on the C-Tetrominoes dataset under the SST setting. Since the model is trained without label supervision, we utilize the *mask matching* approach from Dittadi et al. (2021) to assign semantic pseudo-labels (i.e., *shape, position, color*) to each slot. For the continual learning analysis, we define *task pairs*, denoted as (E$i$/T$j$), which represent the slot representations obtained by evaluating on task $\mathcal{T}_i$ using a model trained sequentially up to task $\mathcal{T}_j$. We visualize these representations using t-SNE (Van der Maaten & Hinton, 2008), where each point corresponds to a slot and is color-coded by its assigned label. Note that we only consider slots corresponding to foreground objects in the following sections. A detailed illustration of this analysis setup is provided in Appendix A.1.

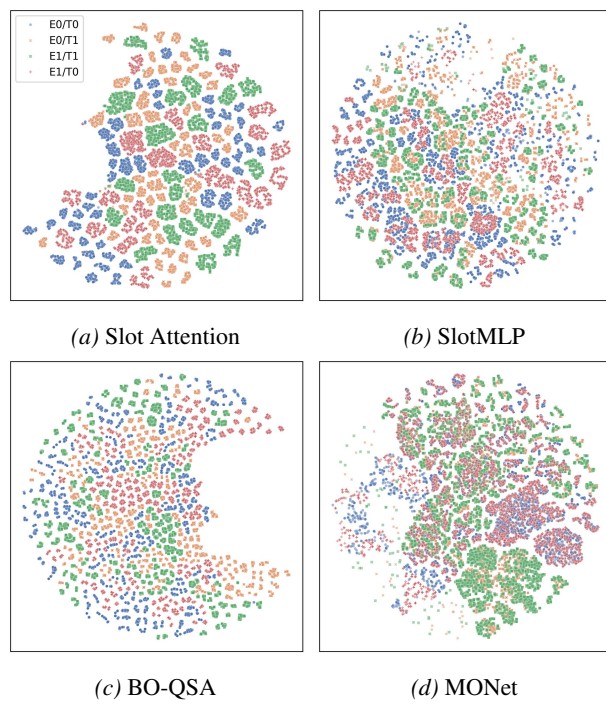

*(a)* Slot Attention      *(b)* SlotMLP

*(c)* BO-QSA      *(d)* MONet

*Figure 1.* Slot inter-task separation visualization via t-SNE.

### 4.2. Factor-Wise Homogeneity for Inter-task Separation

For comparative analysis, we first employ SlotMLP (Locatello et al., 2020), a variant that replaces the Slot Attention module with an MLP, thereby isolating the iterative attention-based binding mechanism to examine its intrinsic effects on representation learning. Additionally, we compare standard Slot Attention with MONet (Burgess et al., 2019), a representative OCL model, and BO-QSA (Jia et al.), an advanced variant of Slot Attention employing learnable queries and bi-level optimization.

**Dynamics of Slot Representation Space.** Fig. 1a highlights several distinct behaviors regarding organization of the latent space. **(1) Local Separation within Single Task.** First, we examine slots within a single task context (e.g., E0/T0), where continual learning effects are absent. In these *intra-task* scenarios, we observe that the representation space naturally organizes into small, localized regions. This pattern is consistent across tasks, indicating that the Slot Attention mechanism inherently induces a clustered structure within the latent space. **(2) Separation from Previous Tasks.** Second, under continual updates, we observe clear *inter-task separation* . Notably, after training on the novel task (T1), distinction between previous (E0/T1) and novel task representations (E1/T1) demonstrates that the model effectively maintains boundaries between old and new knowledge. This suggests that the latent space remains disjoint rather than overlapping, thereby mitigating catastrophic interference. **(3) Separation from Upcoming Tasks with Unseen Factors.** Finally, even with unseen factors, the

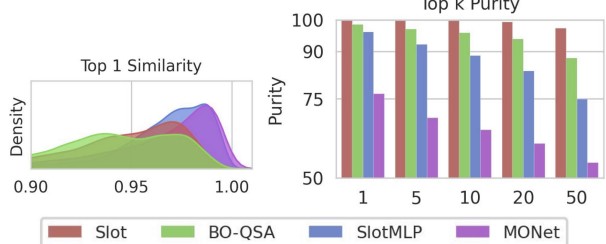

*Figure 2.* Evaluation of inter-task separation using (left) Eq. (2) (right) Eq. (3). "Slot" denotes Slot Attention, which exhibits well-separated features across tasks.

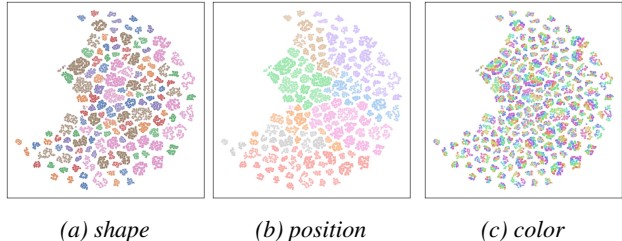

*(a) shape*     *(b) position*     *(c) color*

*Figure 3.* t-SNE visualization of factor-wise homogeneity of Slot Attention. Each color indicates a state of semantic factors.

upcoming task (E1/T0) exhibits no overlap with the current task (E0/T0). This demonstrates that inputs deviating from existing knowledge are naturally mapped to distinct regions of the latent space, rather than merging with previously learned representations.

**Evaluation of Inter-Task Separation.** To quantify separation, we utilize two metrics based on Cosine Similarity. (1) Top-k Inter-Task Class Similarity (Eq. (2)) calculates the average similarity to nearest neighbors from *different* tasks, reflecting inter-task feature overlap for given slot. (2) Top-k Nearest Neighbor Class Purity (Eq. (3)) measures the proportion of neighbors sharing the same task label, reflecting local intra-task consistency.

$$\text{Similarity}^k(s_i) = \frac{1}{k} \sum_{s_j \in \mathcal{N}_k^*(s_i)} \cos(s_i, s_j), \qquad (2)$$

$$\text{Purity}^k(s_i) = \frac{1}{k} \sum_{s_j \in \mathcal{N}_k(s_i)} \mathbb{1}[\tau(s_i) = \tau(s_j)], \qquad (3)$$

where $\mathcal{N}_k(s_i)$ is the set of $k$-nearest neighbors of $s_i$ in the entire latent space $\mathcal{S}$, and $\mathcal{N}_k^*(s_i)$ denotes the $k$-nearest neighbors retrieved from the subset of samples excluding the current task (i.e., $\{s_j \in \mathcal{S} \mid \tau(s_j) \neq \tau(s_i)\}$).

Quantitatively, Fig. 2 demonstrates that Slot Attention yields substantially higher inter-task separation, evidenced by the minimal density of high-similarity neighbors (left), and sustains robust purity even with increasing $k$ (right), maintaining well-separated representations across tasks. We present the evaluation results for longer task sequences in Table 13.

**Factor-wise Homogeneity of Slot Representation** To investigate the underlying mechanism of the observed separation, we visualize slot representations with their semantic labels. As shown in Fig. 3, slots sharing identical semantic factors consistently cluster together (*intra-region consistency*), whereas slots associated with distinct factors are mapped to disjoint regions. By organizing representations based on semantic identity, the model naturally establishes *inter-task separation* between the latent spaces of different tasks ($\mathcal{R}_t$). We refer to this property as **Factor-Wise Homogeneity**: across the slot representation space, slots exhibit semantic consistency with respect to individual factors while remaining well separated from those of different factors. This phenomenon persists across both unseen and previously trained tasks (e.g., E0/T0 and E1/T0), demonstrating that factor-wise homogeneity serves as a robust inductive bias that maintains the structural integrity of task separation throughout the continual learning process.

**Uniqueness of Factor-wise Homogeneity.** We investigate whether factor-wise homogeneity is a generic feature of object-centric models or specific to Slot Attention. In Fig. 2, MONet quantitatively lacks this property, exhibiting high cross-task similarity and poor class purity. In contrast, BO-QSA, advanced variant of Slot Attention, maintains high inter-task separation comparable to the original Slot Attention. These results, supported by task-wise visualizations Fig. 1c and Fig. 1d (semantics-wise visualizations in Fig. 7 and Fig. 8), confirm that factor-wise homogeneity is not a general OCL characteristic but a unique property intrinsic to the Slot Attention mechanism.

### 4.3. Leveraging Factor-Wise Homogeneity for COCL

Building upon our identification of *Factor-wise Homogeneity*, we demonstrate that this intrinsic inter-task separation is a key mechanism to prevent forgetting in COCL

**Benefits of Factor-Wise Homogeneity to COCL.** Crucially, factor-wise homogeneity establishes an intrinsic inter-task separation within the latent space (Fig. 1a). This separation provides a robust inductive bias that mitigates destructive interference between sequential tasks, which often leads to catastrophic forgetting (Buzzega et al., 2020a; Dittadi et al., 2021; Zhu et al., 2021; Kim et al., 2024b). To validate this, we conduct linear probing on frozen slots using a 3-layer MLP to assess representation quality (details in Appendix A.3). After training on sequential tasks, we evaluate: (1) *Task Prediction* (classifying *task pairs* of each slot) and (2) *Semantic Prediction* (classifying semantics of objects from both past and current task). As shown in Table 1, models exhibiting factor-wise homogeneity (i.e., Slot Attention, BO-QSA) achieve high performance in both metrics, whereas models lacking this property degrade significantly. This shows that the observed separation is not

*Table 1.* Impact of inter-task separation on downstream tasks of frozen slots trained on *C-Tetrominoes SST*. We evaluate two tasks: (1) Task Prediction (classifying task pairs) and (2) Semantic Prediction (classifying shape). Throughout the paper, gray shading highlights key results, SA$^{\dagger}$ denotes Slot Attention, and $^{*}$ marks models exhibiting factor-wise homogeneity.

| Model | Task Prediction | | Semantic Prediction | |
|---|---|---|---|---|
| | Acc. | F1 | Acc. | F1 |
| SlotMLP | 77.08 | 76.95 | 74.75 | 62.15 |
| MONet | 76.38 | 75.47 | 61.39 | 42.70 |
| BO-QSA$^{*}$ | 94.93 | 94.71 | 92.72 | 91.39 |
| SA$^{\dagger *}$ | **96.82** | **96.60** | **99.82** | **99.83** |

*Table 2.* Impact of DPR on various OCL models. We report FG-ARI$_{\pm\text{std.}}$ results of various models with and without DPR on *C-Tetrominoes SST*. Parentheses and colors highlights gains of applying DPR.

| Model | E0 / T0 | E0 / T1 | |
|---|---|---|---|
| | | w/o DPR | w/ DPR |
| SlotMLP | $75.83_{\pm.13}$ | $32.85_{\pm.07}$ | $45.75_{\pm.10}$(+13.) |
| MONet | $84.63_{\pm.11}$ | $43.51_{\pm.06}$ | $44.41_{\pm.04}$(+1.) |
| BO-QSA$^{*}$ | $99.98_{\pm.00}$ | $50.46_{\pm.08}$ | **$96.54_{\pm.02}$**(+48.) |
| SA$^{\dagger *}$ | $99.86_{\pm.00}$ | $41.39_{\pm.03}$ | **$98.81_{\pm.00}$**(+57.) |

merely geometric but a functional representation that effectively preserves task boundaries and discriminative semantic features. By ensuring that past knowledge remains linearly retrievable without interference, factor-wise homogeneity serves as a critical foundation for successful COCL.

**Bottleneck in Exploiting Inter-Task Separation at the Decoder.** While OCL benefits generalization (Dittadi et al., 2021), the structural robustness of factor-wise homogeneity alone is insufficient for COCL. A critical limitation lies in the decoder, which acts as a bottleneck in exploiting this inter-task separation. Specifically, while the encoder preserves task-separated slot spaces, the decoder inevitably overfits to the visual statistics of the current task $\mathcal{T}_t$ (Lesort et al., 2019; Montero et al., 2021). This limitation is empirically confirmed in Table 2, where we observe severe degradation on previous tasks (E0/T1) despite well-organized and task-separated slots. To address this without disrupting the established latent structure, we employ a replay strategy that selectively fine-tunes the decoder.

**Leveraging Intrinsic Separation via Minimal Strategy.** We aim to fully exploit the factor-wise homogeneity of Slot Attention in COCL, while avoiding degradation caused by unintended training biases. To demonstrate the significance of this property and overcome the bottleneck of the decoder, we employ a minimal yet effective strategy: *Decoder-only Post Replay*. DPR is founded on two core components: (1) Partial Freezing: We freeze the encoder and Slot Attention module, updating only the decoder. This constraint preserves the intrinsic factor-wise separation observed in the

slot representation space $\mathcal{S}_t$. (2) Post Replay (PR): We introduce a replay phase that occurs only after the initial training on the current task $\mathcal{T}_t$ (performed without replay). By decoupling task adaptation from knowledge preservation, we eliminate interference arising from replay biases during the learning of new tasks (Wu et al., 2019; Rolnick et al., 2019).

**The Critical Role of Latent Space Organization.** We empirically verify whether a well-organized latent space is a necessary condition for effective continual learning. To this end, we apply DPR to models lacking factor-wise homogeneity, specifically SlotMLP and MONet. As shown in Table 2, these models exhibit negligible performance gains (E0/T1) even with replay. In contrast, models preserving valid factor-wise homogeneity (Slot Attention and BO-QSA) achieve substantial performance restoration. This highlights that DPR serves not as a trivial solution, but as a mechanism relying on factor-wise homogeneity, identifying the well-separated latent space as the essential foundation for mitigating catastrophic forgetting.

# 5. Experiments

In this section, we validate the effectiveness of factor-wise homogeneity as a key inductive bias for COCL via DPR. We provide broad experimental validation on both proposed benchmarks and more practical environments, along with an in-depth analysis of this property.

**Training Settings.** DPR maintains a replay buffer of randomly sampled reconstructions from previous tasks (Wu et al., 2018). During the Post Replay phase, the decoder is fine-tuned on a balanced mixture of these replay samples and current task data. Figure 23 illustrates performance across varying buffer sizes. The results indicate that performance generally scales with the buffer size. Table 29 confirms that this trend remains consistent on longer sequences ($|\mathcal{T}|$=6). We selected a default buffer size of 2,000 for all experiments as it provides a reasonable performance. We use the Adam optimizer ($lr$=$4 \times 10^{-4}$) with a 2% warm-up, training for 200 epochs on the initial task, 100 on subsequent tasks, and 50 for replay. Algorithm and detailed configurations of our implementation are provided in Algorithm 1 and Appendix B.3. We also provide discussion of computation overheads in Appendix B.2, where DPR demonstrates superior efficiency compared to standard experience replay using reservoir sampling methods that require replay throughout the training process.

## 5.1. Performance of Factor-wise Homogeneity with DPR

To empirically validate the factor-wise homogeneity property with DPR, we evaluate performance on the two C-OCL benchmarks (Section 3). Table 3 reports the quantitative results. While baselines suffer from degradation on previously learned tasks, applying DPR yields significant performance

*Table 3.* Effectiveness of factor-wise homogeneity and DPR across C-OCL benchmarks. We report FG-ARI$_{\pm\text{std.}}$ results of evaluation on tasks $\mathcal{T}_i$ (E$i$) after sequentially training on novel task $\mathcal{T}_j$ (T$j$).

| Model | E0 / T0 | E0 / T1 | E1 / T1 |
|---|---|---|---|
| *Dataset: C-Tetrominoes SST* | | | |
| SA$^\dagger$(Joint) | | 99.89$_{\pm.00}$ | 99.85$_{\pm.00}$ |
| SA$^\dagger$ | 99.86$_{\pm.00}$ | 41.71$_{\pm.03}$ | 99.79$_{\pm.00}$ |
| SA$^\dagger$+ DPR | 99.88$_{\pm.00}$ | **98.81**$_{\pm.00(+57.)}$ | 99.70$_{\pm.00}$ |
| *Dataset: C-CLEVR SST* | | | |
| SA$^\dagger$(Joint) | | 96.24$_{\pm.01}$ | 96.66$_{\pm.01}$ |
| SA$^\dagger$ | 96.01$_{\pm.01}$ | 63.83$_{\pm.03}$ | 96.21$_{\pm.01}$ |
| SA$^\dagger$+ DPR | 96.03$_{\pm.01}$ | **92.00**$_{\pm.01(+28.)}$ | 95.36$_{\pm.01}$ |

*Table 4.* Consistent effectiveness of factor-wise homogeneity and DPR across various validation scenarios.

| Model | E0 / T0 | E0 / T1 | E1 / T1 |
|---|---|---|---|
| *Dataset: C-Tetrominoes SST-(color)* | | | |
| SA$^\dagger$ | 99.88$_{\pm.00}$ | 70.45$_{\pm.06}$ | 96.85$_{\pm.05}$ |
| SA$^\dagger$+ DPR | 99.95$_{\pm.01}$ | **96.84**$_{\pm.00(+26.)}$ | 99.42$_{\pm.00}$ |
| *Dataset: C-Tetrominoes SST-(shape+color)* | | | |
| SA$^\dagger$ | 99.94$_{\pm.00}$ | 48.03$_{\pm.01}$ | 99.90$_{\pm.00}$ |
| SA$^\dagger$+ DPR | 99.95$_{\pm.00}$ | **94.19**$_{\pm.02(+46.)}$ | 99.51$_{\pm.00}$ |

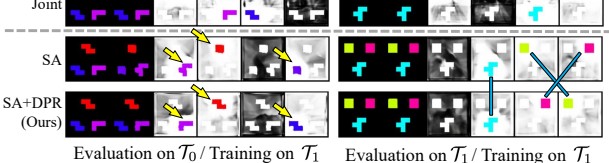

*Figure 4.* Qualitative comparison with and without DPR. DPR successfully recovers the decoding quality on past task (Yellow arrows) without compromising quality on novel tasks (Blue lines).

gains on past tasks without compromising the current task performance across all datasets. This consistency extends to additional scenarios and metrics (Appendix A.4). We further verify that the benefits of DPR extend beyond the *shape* factor to other semantics by introducing tasks involving a novel *color* factor and the compositional combination of both *shape* and *color*. As shown in Table 4, DPR demonstrates consistent improvements across diverse settings, confirming its robustness regardless of the semantic factors. Collectively, these results show that DPR successfully exploits the inherent factor-wise homogeneity of Slot Attention.

Fig. 4 presents reconstruction results for each slot. Without DPR, Slot Attention (middle row) tends to reconstruct objects biased toward the task $\mathcal{T}_1$, generating incorrect shapes. With DPR (bottom row), however, each slot yields semantically accurate object shapes for previous task (right, yellow arrow highlights)), while effectively preserving reconstruction quality in novel tasks (left, blue line highlights). Results for other scenarios are in Appendix A.5.

*Table 5.* Effectiveness of DPR on real-world images. We evaluate on $\mathcal{T}_0$ (E0) after training on $\mathcal{T}_0$ (T0, Initial) and after continuous training up to $\mathcal{T}_3$ (T3, Final) across $|\mathcal{T}| = 4$ tasks (mean$_{\pm\text{std}}$). DPR$^{\dagger\dagger}$ denotes DPR with DINOSAUR.

| Dataset | Model | E0 / T0 | | E0 / T3 | |
|---|---|---|---|---|---|
| | | FG-ARI | mBO$^c$/mBO$^i$ | FG-ARI | mBO$^c$/mBO$^i$ |
| COCO | DINOSAUR | $23.01_{\pm0.8}$ | $45.87_{\pm0.4}$ / $35.43_{\pm0.4}$ | $21.25_{\pm0.6}$ | $42.14_{\pm0.6}$ / $32.76_{\pm0.6}$ |
| | DPR$^{\dagger\dagger}$ | | | $\mathbf{23.03}_{\pm0.8}$ | $\mathbf{43.47}_{\pm0.5}$ / $\mathbf{33.80}_{\pm0.6}$ |
| PASCAL | DINOSAUR | $18.01_{\pm0.6}$ | $55.06_{\pm0.8}$ / $50.11_{\pm0.4}$ | $14.63_{\pm0.7}$ | $51.47_{\pm0.8}$ / $47.26_{\pm0.4}$ |
| | DPR$^{\dagger\dagger}$ | | | $\mathbf{16.69}_{\pm0.6}$ | $\mathbf{52.92}_{\pm0.7}$ / $\mathbf{48.56}_{\pm0.5}$ |

*Table 6.* Performance on real-world scenarios with distribution shifts. We report FG-ARI$_{\pm\text{std.}}$ results.

| Model | E0 / T0 | E0 / T1 | E1 / T1 |
|---|---|---|---|
| | *CLEVR* | *CLEVR* | *CLEVR-Tex* |
| SA$^\dagger$ | $96.51_{\pm.01}$ | $68.68_{\pm.06}$ | $52.75_{\pm.02}$ |
| SA$^\dagger$+ DPR | $96.52_{\pm.01}$ | $\mathbf{82.75}_{\pm.02(+14.)}$ | $51.71_{\pm.03}$ |
| | *Tetrominoes* | *CLEVR* | *CLEVR* |
| SA$^\dagger$ | $99.17_{\pm.01}$ | $37.46_{\pm.01}$ | $97.36_{\pm.00}$ |
| SA$^\dagger$+ DPR | $99.17_{\pm.01}$ | $\mathbf{76.69}_{\pm.00(+39.)}$ | $96.34_{\pm.00}$ |

*Table 7.* Object discovery performance on open-world task ($\mathcal{T}_{t+1}$). We report FG-ARI$_{\pm\text{std.}}$ results. Note that FWH refers to factor-wise homogeneity. Unseen objects from unseen-task $\mathcal{T}_{t+1}$ shows significant performance gap depending on the presence of FWH.

| Model | FWH | DPR | C-Tetrominoes | C-CLEVR |
|---|---|---|---|---|
| SA$^\dagger$ | O | X | $54.49_{\pm.01}$ | $58.68_{\pm.03}$ |
| SA$^\dagger$ | O | O | $\mathbf{98.70}_{\pm.04(+44.2)}$ | $\mathbf{90.72}_{\pm.05(+32.0)}$ |
| Monet | X | X | $53.57_{\pm.05}$ | $58.76_{\pm.09}$ |
| Monet | X | O | $55.00_{\pm.06(+1.4)}$ | $61.36_{\pm.04(+2.6)}$ |

*Table 8.* Linear probing performance on open-world task ($\mathcal{T}_{t+1}$). Advantages of factor-wise homogeneity (well-separated latents) persist across unseen objects.

| Model | FHW | Task Prediction | | Semantic Prediction | |
|---|---|---|---|---|---|
| | | Acc. | F1 | Acc. | F1 |
| MONet | X | 72.37 | 71.87 | 60.37 | 60.97 |
| SA$^\dagger$ | O | **92.36** | **90.14** | **88.72** | **87.95** |

## 5.2. Extension to More Practical Environments

**Real-World Datasets with Pre-trained models.** We extend our evaluation to real-world environments to assess the robustness and scalability of our approach beyond simplified synthetic benchmarks. We utilize COCO (Lin et al., 2014) and PASCAL VOC (Everingham et al., 2010) datasets, creating continual learning scenarios by splitting them into multiple tasks ($|\mathcal{T}| = 2, 4$) with disjoint objects (Appendix A.10). To address the limitations of standard Slot Attention in handling complex images, we employ DINOSAUR (Seitzer et al.), a Slot Attention based model integrated with a pre-trained DINO encoder (Caron et al., 2021).

Table 5 presents the quantitative results on sequential real-world tasks ($|\mathcal{T}| = 4$), reporting FG-ARI and mean Best Overlap (mBO, Eq. (15)) for both instance (i) and class (c) levels. DPR demonstrates consistent performance improvements across all metrics. Qualitatively, t-SNE visualizations confirm that factor-wise homogeneity is preserved, with slots remaining well-separated across tasks (Fig. 21), which shows improved mask prediction quality (Fig. 22). Notably, the performance gap between the baseline and DPR widens in longer task sequences (Table 25). This highlights that DPR effectively leverages the preserved factor-wise homogeneity to mitigate forgetting, providing a robust inductive bias even in long-term continual learning scenarios.

**Generalization to Real-World Scenarios.** To validate robustness against textural shifts, we evaluate performance on the CLEVR-Tex (Karazija et al.) dataset. We report retention on the initial task $\mathcal{T}_0$ (CLEVR) following training on the texture-rich $\mathcal{T}_1$ (CLEVR-Tex). As shown in Table 6 (Top), the consistent performance gains confirm that our findings hold even in challenging settings with complex textures. We extend our evaluation to cross-datasets (Tetrominoes to CLEVR) where there are shifts in appearance (2D VS 3D), background (black VS grey), lighting (X VS O). Despite these distribution shifts, results from Table 6 (Bottom) indicate that our findings remains valid, enabling the frozen encoder to provide robust latent representations that generalize across diverse object types from past tasks.

Furthermore, we analyzed the scalability of our findings to *Open-world* (Bendale & Boult, 2015) by evaluating the model on an unseen task $\mathcal{T}_{t+1}$ after freezing the factor-wise homogeneity encoder at task $\mathcal{T}_t$. To address our claim that the decoder (or classifier) acts as a bottleneck in exploiting factor-wise homogeneity, we updated only the decoder using a minimal sample set from $\mathcal{T}_0, ..., \mathcal{T}_t, \mathcal{T}_{t+1}$ to alleviate this bottleneck. This validates factor-wise homogeneity in a *partially* open-world scenario while keeping the encoder strictly agnostic to the $\mathcal{T}_{t+1}$. Our results on Table 7 (object discovery) and Table 8 (representation analysis) confirms the scalability and robust potential of factor-wise homogeneity in partially open-world scenario.

## 5.3. Ablation Study

**Ablation of the Decoder.** Motivated by the robust inter-task separation observed in slots, we investigate whether

*Table 9.* Impact of different mask predictions. FG-ARI$_{\pm\text{std.}}$ results using decoder-based (*Dec.*) and slot attention (*Attn.*) masks.

| Model | Mask | Dataset: *C-CLEVR SST* | |
|---|---|---|---|
| | | E0 / T0 | E0 / T1 |
| SA$^\dagger$ | *Dec.* | $99.86_{\pm.00}$ | $41.71_{\pm.03}$ |
| SA$^\dagger$ | *Attn.* | $92.54_{\pm.05}(-7.)$ | $85.19_{\pm.14}(+43.)$ |
| SA$^\dagger$+DPR | *Dec.* | $99.88_{\pm.00}(+0.)$ | $\mathbf{98.81}_{\pm.00}(+57.)$ |

*Table 10.* Comparison and combination of factor-wise homogeneity with diffusion generative replays. We report FG-ARI$_{\pm\text{std.}}$ results evaluated on the COCO dataset. RS denotes reservoir sampling, where the experience replay method randomly stores samples in memory utilizing a reservoir sampling strategy.

| Method | Memory | Replay | FG-ARI | |
|---|---|---|---|---|
| | | | E0 / T0 | E3 / T1 |
| DINOSAUR | - | - | $23.54_{\pm.03}$ | $22.27_{\pm.02}$ |
| DDGR | DDGR | RS | | $24.08_{\pm.04}(1.80)$ |
| DPR | Random | DPR | $23.54_{\pm.03}$ | $23.63_{\pm.01}(1.35)$ |
| DPR$^\dagger$ | DDGR | DPR | | $24.03_{\pm.01}(1.76)$ |
| DPR$^\ddagger$ | DDGR | RS+DPR | | $\mathbf{24.25}_{\pm.05}(1.98)$ |

utilizing raw attention masks can serve as an alternative for object discovery, potentially bypassing the bottleneck of the decoder. Table 9 confirms that while slot attention masks are highly robust against continual updates—aligning with our central finding of factor-wise homogeneity—they lack the segmentation precision of decoder outputs in complex environments (e.g., C-CLEVR) (Seitzer et al.; Kakogeorgiou et al., 2024). This demonstrates that the decoder is still required for high-quality object discovery, showing the necessity of our DPR strategy to recover the performance.

**Ablation and Comparison of the Replay Strategies.** We conduct an ablation study on the replay strategy by evaluating Post Replay (PR), which relaxes the *decoder-only* constraint of DPR by updating the full model. We compare these approaches against standard baselines: Experience Replay (ER) (Lopez-Paz & Ranzato, 2017a; Rebuffi et al., 2017; Chaudhry et al., 2019; Buzzega et al., 2020a) and Generative Replay (GR) (Shin et al., 2017; Wu et al., 2018; Ayub & Wagner; Zhai et al., 2019; Gao & Liu, 2023). The results suggest that while DPR provides a strict and minimal implementation ensuring factor-wise homogeneity, further gains are achievable via PR, provided that this homogeneity remains intact (Table 19 in Appendix A.6). Notably, both DPR and PR demonstrate performance comparable to ER, even though ER integrates replay samples directly during the novel task training phase.

We further evaluate the effectiveness of factor-wise homogeneity by comparing with DDGR (Gao & Liu, 2023). DDGR utilizes diffusion models to generate replay samples instead of memorizing raw data from past tasks, performing experience replay with reservoir sampling. We modified

DDGR to use multiple object class labels as conditional inputs of the diffusion models to generate multi-object replay samples. While we initially compare DDGR with DPR, we emphasize that DDGR focuses on 'which sample' to save (coreset selection), whereas DPR addresses 'how' to replay them. This distinction makes them inherently compatible rather than mutually exclusive. Consequently, we utilize diffusion generated replay samples for DPR, denoting DPR using diffusion-generated samples as DPR$^\dagger$, and the combination of them as DPR$^\ddagger$. The results on the COCO dataset are presented in Table 10. We found that DPR with diffusion-generated samples is comparable to DDGR, despite not requiring replay throughout the entire training process, thereby maintaining superior efficiency (Table 27). Furthermore, applying DPR as a modular plugin to DDGR further enhances performance, achieving the best results. These findings also indicate that DPR is robust in buffer-free settings via generative diffusion, obviating the need for a physical buffer in data-sensitive scenarios where privacy is a critical consideration.

Additionally, we provide a comparative analysis with continual regularization methods, demonstrating both the effectiveness of DPR and its robustness when integrated with these regularizer (Table 20 in Appendix A.8).

### 5.4. What Drives Factor-wise Homogeneity and Inter-task Separation?

In this section, we investigate which module drives factor-wise homogeneity and inter-task separation, identifying the GRU as the key mechanism driving these phenomena.

**Module Influence on Slot Organization.** Let $\mathbf{h}_m : \mathbb{R}^D \to \mathbb{R}^D$ denote the function of a specific module $m$ (i.e., Cross-Attention, GRU, or MLP) applied to the slot representation $\mathbf{s} \in \mathbb{R}^D$. We first investigate the influence of each module on slots. By performing Singular Value Decomposition (SVD) on the Jacobian matrix (Eq. (4)) (Novak et al., 2018), we isolate and interpret the contribution of each module through its singular vectors. Specifically, we utilize: (1) The Frobenius Norm of the Jacobian to quantify the intensity of the update dynamics, indicating how strongly the module modifies the incoming information. (2) The Effective Rank (Roy & Vetterli, 2007) to assess the richness of the representation, providing empirical evidence that the model processes information within a sufficiently high-dimensional latent space.

$$\mathbf{J}_{\mathbf{h}_m} = \frac{\partial \mathbf{h}_m(\mathbf{s})}{\partial \mathbf{s}} = \mathbf{U}\mathbf{\Sigma}\mathbf{V}^T \qquad (4)$$

Table 11 highlights the influence of the GRU. While the MLP shows high diversity, its low Frobenius Norm indicates a passive contribution. The GRU, however, combines the highest Frobenius Norm with comparable diversity, suggest-

*Table 11.* Analysis of the influence of Slot Attention modules . We report the Frobenius norm and Effective Rank for (1) Attention, (2) GRU, and (3) MLP.

|  | Attention | GRU | MLP |
|---|---|---|---|
| Norm | 1131.06 | 1380.01 | 761.01 |
| Rank ($D = 64$) | 23.91 | 31.17 | 34.21 |

*Table 12.* Consistency of GRU's slow mode ranks. GRU exhibits consistent Spearman's rank correlations $\rho$ across objects with identical semantics (*shape*, *position*, *color*).

| (E$i$)/(T$j$) | Identical | Non-Identical |
|---|---|---|
| E0 / T0 | 0.9921 | 0.7931 |
| E1 / T1 | 0.9920 | 0.7958 |
| E0 / T1 | 0.9935 | 0.8054 |
| E0 / T1 VS E1 / T1 | - | 0.7929 |

ing that it serves as the core mechanism for the organization of the latent slot space.

**Object-wise Consistency of GRU.** To investigate the GRU's influence to slots, we base our analysis in the time-scale dynamics of gated RNNs. Prior studies establish that gating mechanisms enable the preservation of long-range information by inducing *slow mode*: stable subspaces where hidden states evolve slowly. Specifically, gates generate marginally stable directions with eigenvalues clustered near one, enabling long-timescale stability and supporting continuous attractors (Tallec & Ollivier, 2018; Krishnamurthy et al., 2022; Can et al., 2020). Furthermore, RNNs operate near identity transformations, driven by a low-dimensional set of dominant, slowly evolving eigen-directions (Smith et al., 2021; Miller & Hardt). Such dynamics naturally give rise to slowly changing hidden-states, providing an inherent mechanism for maintaining global information over time.

We define slow modes within the GRU as hidden dimensions characterized by consistent update gate activations ($z_t$). Low $z_t$ values enforce minimal state transitions, thereby creating stable conduits for semantic persistence. Leveraging the magnitude of $z_t$ as an empirical metric of dimensional stability, we investigate the structural basis of factor-wise homogeneity. Specifically, we compute the Spearman's rank correlation ($\rho$) on hidden dimensions ranked by update gate magnitude to validate the hypothesis that slots encoding identical semantic factors (e.g., *shape*, *position*, *color*) align along the same set of highly stable dimensions. Details are provided in Appendix A.9.

We first verify the alignment between GRU slow modes and semantic factors. As shown in Table 12, objects sharing identical factors exhibit significant correlation ($\rho \approx 0.99$), whereas those with differing factors show a distinctively lower correlation ($\rho \approx 0.80$). This confirms that the GRU performs consistent updates conditioned on semantics, establishing factor-specific slow modes. Crucially, this con-

sistency persists cross tasks in continual learning. Table 12 (bottom) indicates that inter-task separation aligns with the low correlations observed for differing semantic factors, indicating that the GRU preserves factor-specific slow modes despite continual updates. Furthermore, ablation studies show that the GRU's recurrent updates are essential for sustaining this inter-task separation (Appendix A.7).

In summary, we establish the GRU as the essential for both factor-wise homogeneity and inter-task separation. Jacobian analysis confirms that the GRU drives the organization of the latent slot space. Furthermore, via slow modes governed by update gate activations, the GRU structurally aligns identical semantic factors into highly stable dimensions while separating differing factors from sequential tasks, ensuring robust inter-task separation in continual learning.

## 6. Conclusion & Future Work

In this paper, we take the first step forward enabling continual object-centric learning. We highlight the distinctive property of Slot Attention, *factor-wise homogeneity*. This property organizes the slot representations into small, well-separated regions that preserve identical semantics across tasks, establishing an intrinsic inter-task separation. We demonstrate that this provides a crucial inductive bias for mitigating catastrophic forgetting and introduce DPR, a minimal strategy that addresses the bottleneck of the decoder while preserving the robust representations. Extensive experiments on our COCL benchmarks validate that leveraging this factor-wise homogeneity is sufficient for stable continual learning, establishing a fundamental baseline for future COCL research.

Representation collapse between distinctive features remains a fundamental challenge not only in continual learning but in neural network. Our findings suggest that factor-wise homogeneity may offer a promising direction for mitigating such collapse, both for seen and unseen features. We hope future work explores the broader implications of factor-wise homogeneous representations across diverse frameworks and applications.

## Acknowledgments

This work was supported by the National Research Foundation of Korea (NRF) grant funded by the Korea government (MSIT) (No.2022R1A2C2012054, Development of AI for Canonicalized Expression of Trained Hypotheses by Resolving Ambiguity in Various Relation Levels of Representation Learning) (90%) and Institute of Information communications Technology Planning Evaluation (IITP) grant funded by the Korea government (MSIT) (No.2019-0-01842, Artificial Intelligence Graduate School Program (GIST)) (10%).

## Impact Statement

This paper presents work whose goal is to advance the field of Machine Learning by improving the stability and robustness of object-centric representations. As our contribution is primarily methodological, we do not anticipate specific negative societal consequences beyond those generally associated with the development of deep learning models, such as the reliance on potentially biased datasets or the environmental cost of model training. While potential societal impacts ultimately depend on the application and deployment context, we believe that advancing fundamental representation learning mechanisms serves as a stepping stone toward building more reliable and interpretable AI systems.

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

# A. Appendix: Experiment Details and Results

## A.1. Slot Attention Representation Space

We trained the Slot Attention model as described in Appendix B.4.

**Implementation of t-SNE visualization**    We visualize the slot representations using t-SNE (Van der Maaten & Hinton, 2008) based on semantic pseudo-labels: *shape*, *position*, and *color*. These labels are assigned to each slot via *mask matching*, which uses the Hungarian algorithm (Kuhn, 1955), following the procedure in (Dittadi et al., 2021) and the matching loss formulation in (Locatello et al., 2020). Given $M$ ground-truth object masks and $K$ predicted masks for each slot, we treat the label assignment as a bipartite matching problem. We compute a cost matrix between all pairs of predicted and ground-truth masks, and apply the Hungarian algorithm to find an optimal one-to-one assignment that minimizes the cost. Once the matching is established, we transfer the semantic labels of the matched ground-truth objects to the corresponding slots.

**t-SNE visualization of Different Semantic Factors**    Each dot in the visualization is colored according to its semantic pseudo-label. The marker shape of each dot indicates the source of the slot: a *circle* represents slots from (E0/T0), a *cross* represents (E0/T1), *square* represents (E1/T1), *plus* represents (E1/T0).

Fig. 5 shows the t-SNE visualization of Slot Attention slot representations using the C-Tetrominoes SST dataset. We used 2,000 validation images, each contributing $K=4$ slots. Although the background slots are included in the t-SNE computation, they are excluded from the visualization for clarity. Note that we discretized labels for *position* since have continuous values.

The semantic labels are defined as follows:

- *Shape*: 7 foreground object classes + 1 background class

- *Position*: 8 foreground spatial bins + 1 background class

- *Color*: 10 foreground color classes + 1 background class

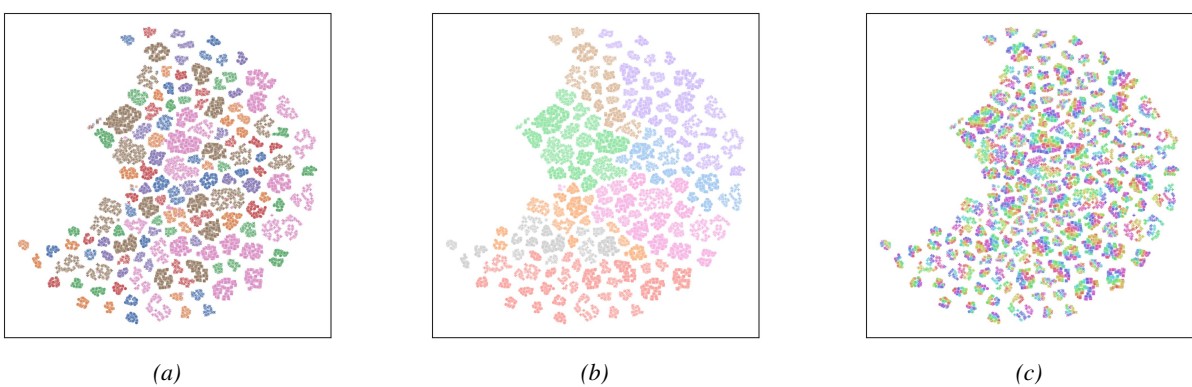

*(a)*                                *(b)*                                *(c)*

*Figure 5.* t-SNE visualization results of slots from Slot Attention. Each color of the dots represent (a) *shape*, (b) *position*, (c) *color*.

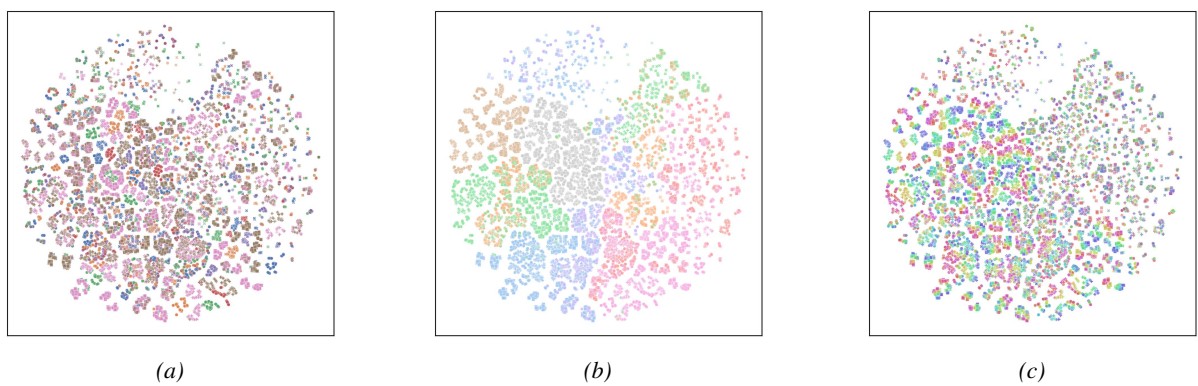

*(a)*          *(b)*          *(c)*

*Figure 6.* t-SNE visualization results of slots from SlotMLP. Each color of the dots represent (a) *shape*, (b) *position*, (c) *color*.

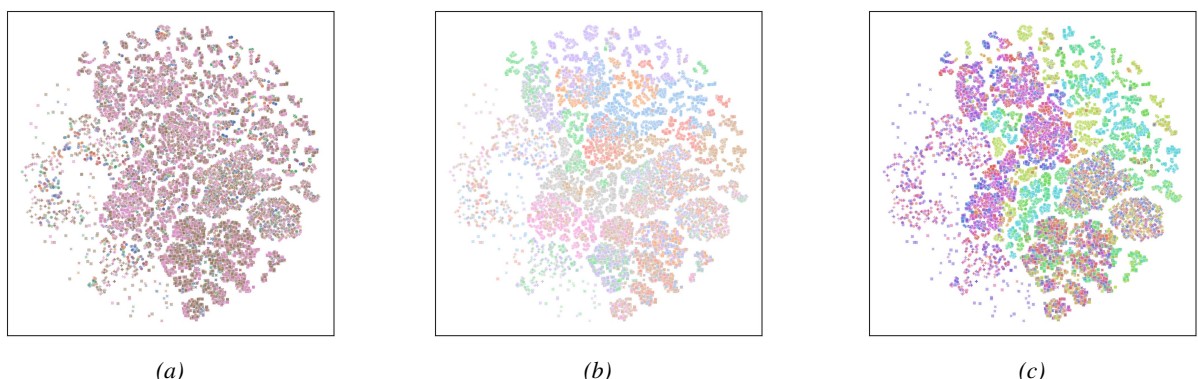

*(a)*          *(b)*          *(c)*

*Figure 7.* t-SNE visualization results of slots from MONet. Each color of the dots represent (a) *shape*, (b) *position*, (c) *color*.

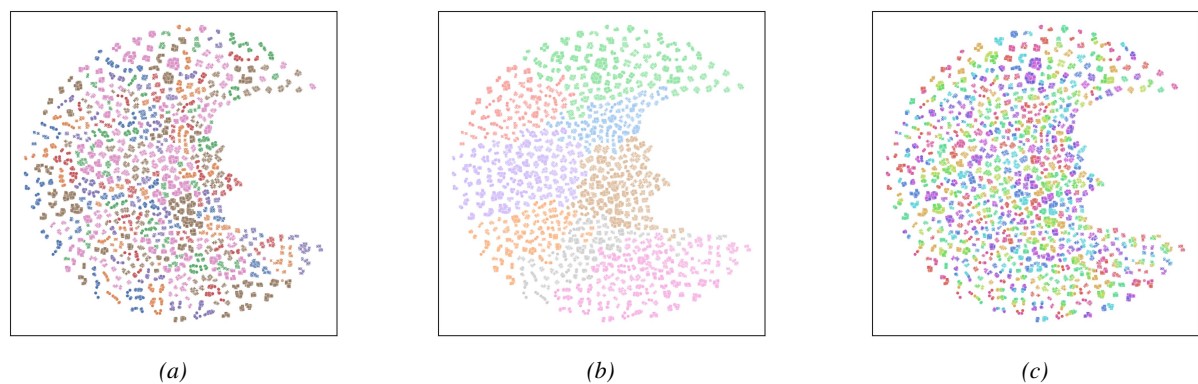

*(a)*          *(b)*          *(c)*

*Figure 8.* t-SNE visualization results of slots from BO-QSA. Each color of the dots represent (a) *shape*, (b) *position*, (c) *color*.

**t-SNE visualization of Inter-task Separation**  To qualitatively analyze the structural evolution of the latent space under continual learning, we employ t-SNE visualization on slot representations collected across different tasks. We introduce a specific notation, (E$i$/T$j$) to precisely describe the evaluation context. (E$i$/T$j$) denotes evaluation on task $\mathcal{T}_i$ using a model continuously trained from the initial task up to task $\mathcal{T}_j$.

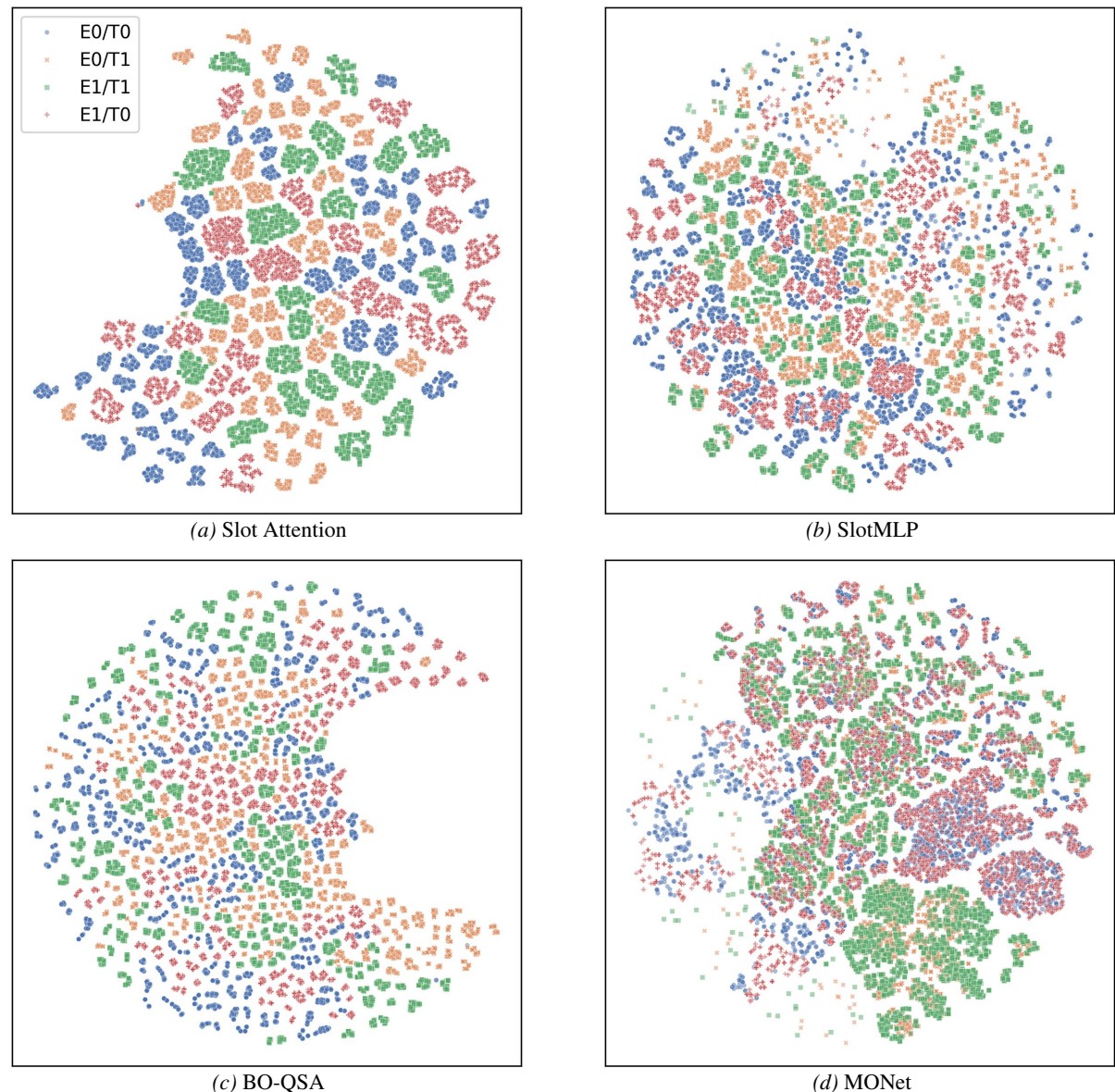

*(a)* Slot Attention

*(b)* SlotMLP

*(c)* BO-QSA

*(d)* MONet

*Figure 9.* Slot inter-task separation visualization via t-SNE.

## A.2. Robustness of Inter-Task Separation against Task Scaling

In Section 4.2, we demonstrate the intrinsic factor-wise homogeneity property of Slot Attention, which manifests as inter-task separation. Figure 2 presents the quantitative evaluation of this property using two metrics (Eq. (2) and Eq. (3)). While the primary evaluation is conducted on the C-Tetrominoes SST dataset, which contains two sequential tasks ($|\mathcal{T}| = 2$), we further validate the generalizability of this inter-task separation over longer task sequences using the C-Tetrominoes MST dataset ($|\mathcal{T}| = 6$).

*Table 13.* Robustness of inter-task separation across extended task sequences. We report the quantified separation metrics (Eq. (3))) evaluated on the C-Tetrominoes SST ($|\mathcal{T}| = 2$) and MST ($|\mathcal{T}| = 6$) datasets.

| Top K | C-Tetrominoes SST ($|\mathcal{T}| = 2$) | C-Tetrominoes MST ($|\mathcal{T}| = 6$) |
|---|---|---|
| 1 | 99.90 | 98.04 |
| 10 | 99.69 | 96.58 |
| 50 | 97.42 | 91.96 |

The risk of collapse as the number of sequential tasks increases is an inherent challenge in continual learning. While some degradation is expected over extended sequences, our results in Table 13 demonstrate that although purity scores slightly decrease, the inter-task separation facilitated by factor-wise homogeneity remains robust even as the sequence scales.

## A.3. Linear Probing on Slots

In this section, we construct linear probing on frozen slots to validate that Slot Attention's intrinsic inter-task separation established by factor-wise homogeneity provides a robust inductive bias that mitigates destructive interference between sequential tasks. After training sequential tasks, we evaluate: (1) *Task Prediction* (classifying *task pairs* of each slot) and (2) *Semantic Prediction* (classifying semantics of objects from both past and current task). Results demonstrates that observed separation is not merely a geometric property, but a functional representation that downstream models can effectively leverage to achieve robust generalization. By ensuring that past knowledge remains linearly retrievable without interference, factor-wise homogeneity serves as a fundamental prerequisite for successful COCL.

**Training Details.** After training on sequential task (*C-Tetrominoes SST*), we training a MLP classifier on top of frozen slots to predict its properties: (1) *task pairs*, (2) semantic factors. We use the pseudo-label used in our empirical analysis (Section 4.1 and Appendix A.1) as an ground-truth labels for each slot. We collect all slot representations using the trained models and datasets, and train MLP classifier (Table 14). We report details of our training hyper-parameters in Table 15

First, *Task Prediction* aims to classify *task pairs* of each slot, measuring quality of inter-task representation separation across sequential tasks. There are four labels of *task pairs* (i.e., E0/T0, E1/T1, E0/T1, E1/T1) representing the slot representations obtained by evaluating on task $\mathcal{T}_i$ using a model trained sequentially up to task $\mathcal{T}_j$ ($E_i/T_j$).

*Semantic Prediction* aims to classify semantic properties of objects from both past and current task. We use pseudo-label Section 4.1 as an ground-truth. However, since our *C-Tetrominoes SST* introduces disjoint and novel *shape* factors during sequential tasks, we perform linear probing on *shape* factors. After training on sequential task (*C-Tetrominoes SST*), we collect slots from two task pairs (E0/T1, E1/T1). Two pairs indicate slots representing object of an specific *shape* factor from previous task ($\mathcal{T}_0$) and object of an specific *shape* factor from current task ($\mathcal{T}_1$) after training on current task ($\mathcal{T}_1$). We identify that whether structured latent space not only preserves distinct task boundaries but also renders semantic features highly discriminative.

*Table 14.* Architecture of the Linear Probing MLP. We employ a simple 3-layer MLP to evaluate the frozen slot representations. $D_{in}$, $H$, and $C$ denote the input slot dimension, hidden dimension, and number of target classes, respectively.

| Layer | Operation | Dimensions | Activation |
|---|---|---|---|
| Layer 1 | Linear | $D_{in} \to H$ | ReLU |
| Layer 2 | Linear | $H \to H$ | ReLU |
| Output | Linear | $H \to C$ | – |

*Table 15.* Hyperparameters for Linear Probing.

| Hyperparameter | Value |
|---|---|
| Optimizer | Adam |
| Peak Learning Rate | $4 \cdot 10^{-4}$ |
| Weight Decay | $5 \cdot 10^{-4}$ |
| Batch Size | 64 |
| Total Training Steps | $50,000$ |
| Learning Rate Scheduler | Linear Warmup (10%) + Cosine Decay |

*Table 16.* Downstream performance($_{\pm \text{std.}}$) of 3-layer MLP on frozen slots using C-Tetrominoes SST. Task Prediction (classifying task pairs) and Semantic Prediction (classifying shape). SA$^\dagger$ denotes Slot Attention, and $^*$ marks models exhibiting factor-wise homogeneity.

| Model | Task Prediction | | Semantic Prediction | |
|---|---|---|---|---|
| | Acc. | F1 | Acc. | F1 |
| SlotMLP | $77.08_{\pm.00}$ | $76.95_{\pm.01}$ | $74.75_{\pm.00}$ | $62.15_{\pm.01}$ |
| MONet | $76.38_{\pm.01}$ | $75.47_{\pm.01}$ | $61.39_{\pm.01}$ | $42.70_{\pm.02}$ |
| BO-QSA$^*$ | $94.93_{\pm.00}$ | $94.71_{\pm.00}$ | $92.72_{\pm.01}$ | $91.39_{\pm.00}$ |
| SA$^{\dagger*}$ | $\mathbf{96.82}_{\pm.00}$ | $\mathbf{96.60}_{\pm.00}$ | $\mathbf{99.82}_{\pm.00}$ | $\mathbf{99.83}_{\pm.00}$ |

**Linear Probing for Property Prediction.** As presented in Table 16, the quantitative results clearly demonstrate the impact of inter-task separation of factor-wise homogeneity on representation quality. Models lacking this property, such as SlotMLP and MONet, exhibit suboptimal performance, particularly in Semantic Prediction where accuracy drops to $61.39\% \sim 74.75\%$. This degradation indicates that their latent representations suffer from severe entanglement, making it difficult for MLP classifiers to retrieve preserved knowledge. In contrast, models exhibiting factor-wise homogeneity (i.e., Slot Attention and BO-QSA) achieve superior performance across all metrics. This empirical evidence confirms that the structured latent space established by factor-wise homogeneity does not merely preserve distinct task boundaries but renders semantic features highly discriminative and immediately accessible. Crucially, this validates that the observed separation acts as a functional representation that downstream networks can effectively leverage to achieve robust generalization, serving as a fundamental prerequisite for successful continual object-centric learning.

## A.4. Validation Results of DPR

To ensure that the performance improvements observed with DPR are statistically reliable and not merely incidental, we report results averaged over five independent runs, including mean and standard deviation. We employ four quantitative metrics for evaluation: (1) FG-ARI (Eq. (10)), (2) MSE (Eq. (11)), (3) mSC (Eq. (13)), and (4) SC (Eq. (12)). Table 17 and 18 present a comparative analysis between the baseline and the model applied with DPR. The results indicate that DPR yields sustained performance gains across various continual learning scenarios. Furthermore, as illustrated in Fig. 10, DPR demonstrates consistent improvements across diverse metrics, demonstrating that it effectively preserves object-centric structures while mitigating forgetting.

*Table 17.* Consistent effectiveness of factor-wise homogeneity and DPR across various validation scenarios. We report FG-ARI$_{\pm\text{std.}}$ results of evaluation on tasks $\mathcal{T}_i$ (E$i$) after sequentially training on novel task $\mathcal{T}_j$ (T$j$).

| Dataset | Model | E0 / T0 | E0 / T1 | E1 / T1 |
|---|---|---|---|---|
| *C-Tetrominoes SST* | SA$^\dagger$ (Joint) | 99.89$_{\pm.00}$ | | 99.85$_{\pm.00}$ |
| | SA$^\dagger$ | 99.86$_{\pm.00}$ | 41.71$_{\pm.03}$ | 99.79$_{\pm.00}$ |
| | SA$^\dagger$ + DPR | 99.88$_{\pm.00}$ | **98.81**$_{\pm.00}$ | 99.70$_{\pm.00}$ |
| *C-Tetrominoes SSM* | SA$^\dagger$ (Joint) | 99.86$_{\pm.00}$ | | 99.82$_{\pm.00}$ |
| | SA$^\dagger$ | 99.86$_{\pm.00}$ | 50.44$_{\pm.04}$ | 99.66$_{\pm.00}$ |
| | SA$^\dagger$ + DPR | 99.86$_{\pm.00}$ | **99.10**$_{\pm.01}$ | 99.42$_{\pm00}$ |
| *C-CLEVR SST* | Joint | 96.24$_{\pm.01}$ | | 96.66$_{\pm.01}$ |
| | SA$^\dagger$ | 96.01$_{\pm.01}$ | 63.83$_{\pm.03}$ | 96.21$_{\pm.01}$ |
| | SA$^\dagger$ + DPR | 96.03$_{\pm.01}$ | **92.00**$_{\pm.01}$ | 95.36$_{\pm.01}$ |
| *C-CLEVR SSM* | Joint | 95.19$_{\pm.02}$ | | 95.11$_{\pm.02}$ |
| | SA$^\dagger$ | 96.02$_{\pm.01}$ | 65.54$_{\pm.04}$ | 93.81$_{\pm.04}$ |
| | SA$^\dagger$ + DPR | 96.02$_{\pm.01}$ | **93.55**$_{\pm.01}$ | 95.81$_{\pm.02}$ |

*Table 18.* Consistent effectiveness of factor-wise homogeneity and DPR on validation scenarios with longer tasks. We report FG-ARI$_{\pm\text{std.}}$ results of evaluation on tasks $\mathcal{T}_i$ (E$i$) after sequentially training on novel task $\mathcal{T}_j$ (T$j$).

| Dataset | Model | E0 / T5 | E1 / T5 | E2 / T5 | E3 / T5 | E4 / T5 | E5 / T5 |
|---|---|---|---|---|---|---|---|
| *C-Tetrominoes MST* | SA$^\dagger$ Joint | 99.84$_{\pm.00}$ | 99.84$_{\pm.00}$ | 99.81$_{\pm.00}$ | 99.84$_{\pm.00}$ | 99.80$_{\pm.00}$ | 99.84$_{\pm.00}$ |
| | SA$^\dagger$ | 49.80$_{\pm.03}$ | 61.15$_{\pm.06}$ | 48.94$_{\pm.08}$ | 36.92$_{\pm.06}$ | 41.70$_{\pm.06}$ | 99.83$_{\pm.00}$ |
| | SA$^\dagger$ + DPR | **96.41**$_{\pm.01}$ | **98.87**$_{\pm.01}$ | **98.48**$_{\pm.00}$ | **98.04**$_{\pm.01}$ | **97.93**$_{\pm.01}$ | 99.10$_{\pm.01}$ |
| *C-Tetrominoes MST* | SA$^\dagger$ Joint | 93.90$_{\pm.00}$ | | 94.08$_{\pm.00}$ | 91.85$_{\pm.00}$ | 93.33$_{\pm.01}$ | 93.33$_{\pm.01}$ |
| | SA$^\dagger$ | 68.12$_{\pm.04}$ | 65.74$_{\pm.04}$ | 66.62$_{\pm.05}$ | 66.12$_{\pm.04}$ | 69.11$_{\pm.04}$ | 95.30$_{\pm.01}$ |
| | SA$^\dagger$ + DPR | **93.05**$_{\pm.01}$ | **86.86**$_{\pm.01}$ | **92.18**$_{\pm.00}$ | **94.54**$_{\pm.01}$ | **88.96**$_{\pm.02}$ | **95.07**$_{\pm.01}$ |

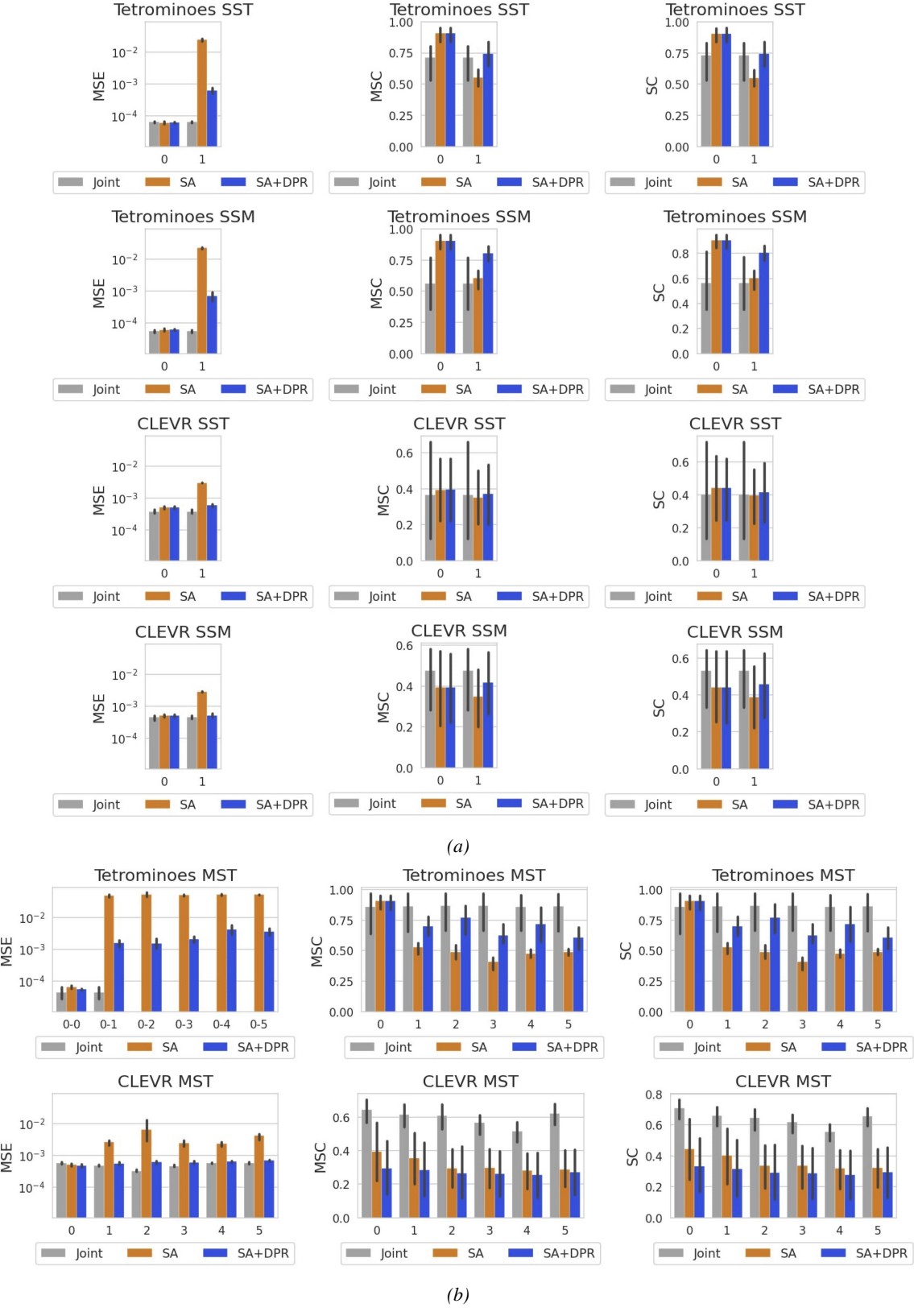

*Figure 10.* Evaluation results of various metrics. (Left) MSE($\pm$std.) ($\downarrow$), (Middle) mSC($\pm$std.) ($\uparrow$), (Right) SC($\pm$std.) ($\uparrow$). Evaluation is performed on previous tasks $\mathcal{T}_i$ (E$i$) evaluated after training on novel task $\mathcal{T}_j$ (T$j$). SA denotes the baseline Slot Attention, which suffers severe degradation, whereas DPR effectively restores performance.

## A.5. Reconstruction Results of Decoder only Post Replay

In this section, we reconstruction results of DPR using three different training scenarios (SST, SSM, MST) compared with vanilla Slot Attention.

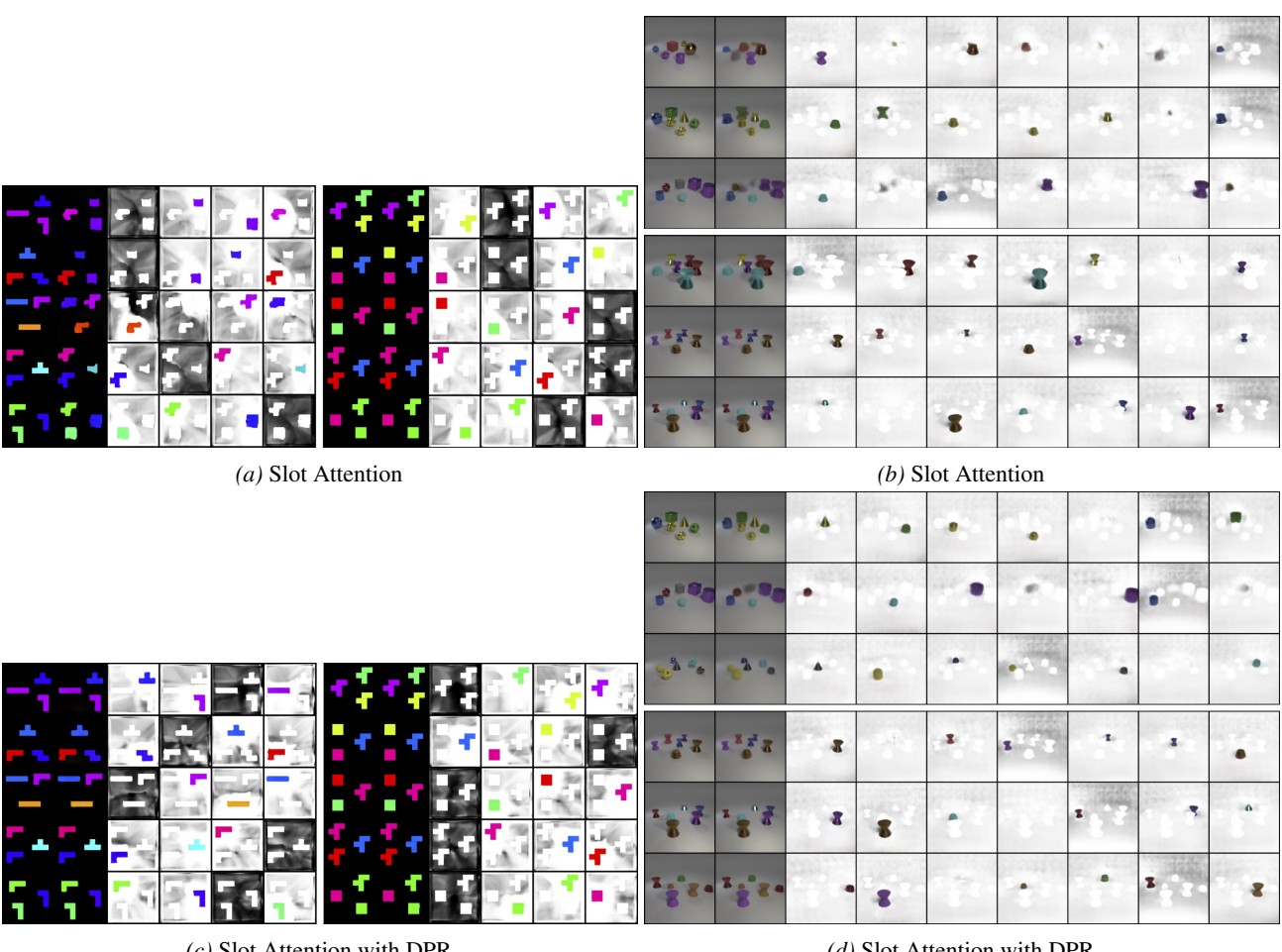

*(a)* Slot Attention       *(b)* Slot Attention

*(c)* Slot Attention with DPR      *(d)* Slot Attention with DPR

*Figure 11.* Reconstruction results of Slot Attention using with (a,b) and without DPR (c,d) on C-Tetrominoes SST, and C-CLEVR SST. We used the model after continuously training the last task ($\mathcal{T}_1$). Figures on left (top) are images of $\mathcal{T}_0$, and figures on right (bottom) are images of $\mathcal{T}_1$

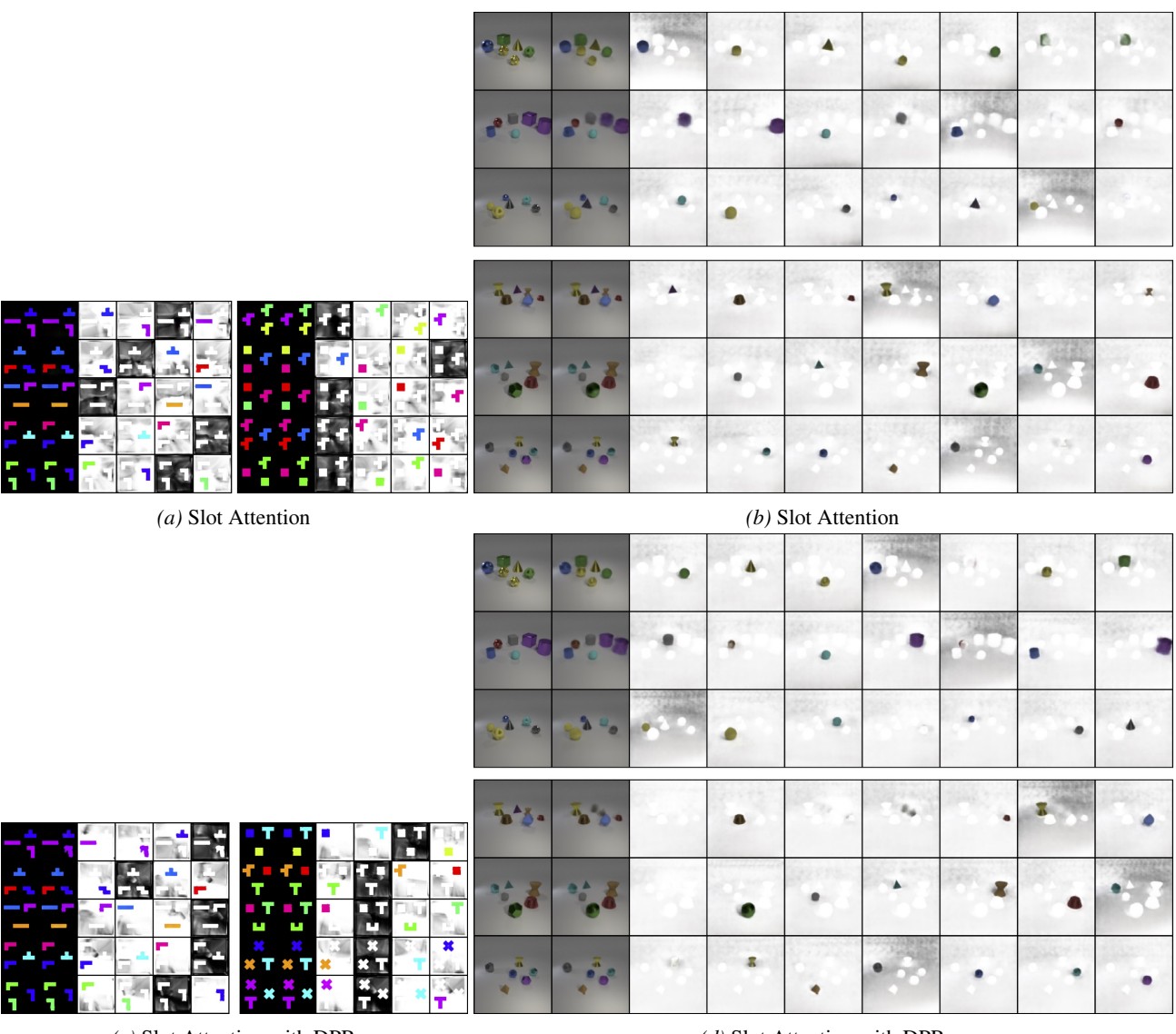

*(a)* Slot Attention

*(b)* Slot Attention

*(c)* Slot Attention with DPR

*(d)* Slot Attention with DPR

*Figure 12.* Reconstruction results of Slot Attention using with (a,b) and without DPR (c,d) on C-Tetrominoes SSM, and C-CLEVR SSM. We used the model after continuously training the last task ($\mathcal{T}_1$). Figures on left (top) are images of $\mathcal{T}_0$, and figures on right (bottom) are images of $\mathcal{T}_1$

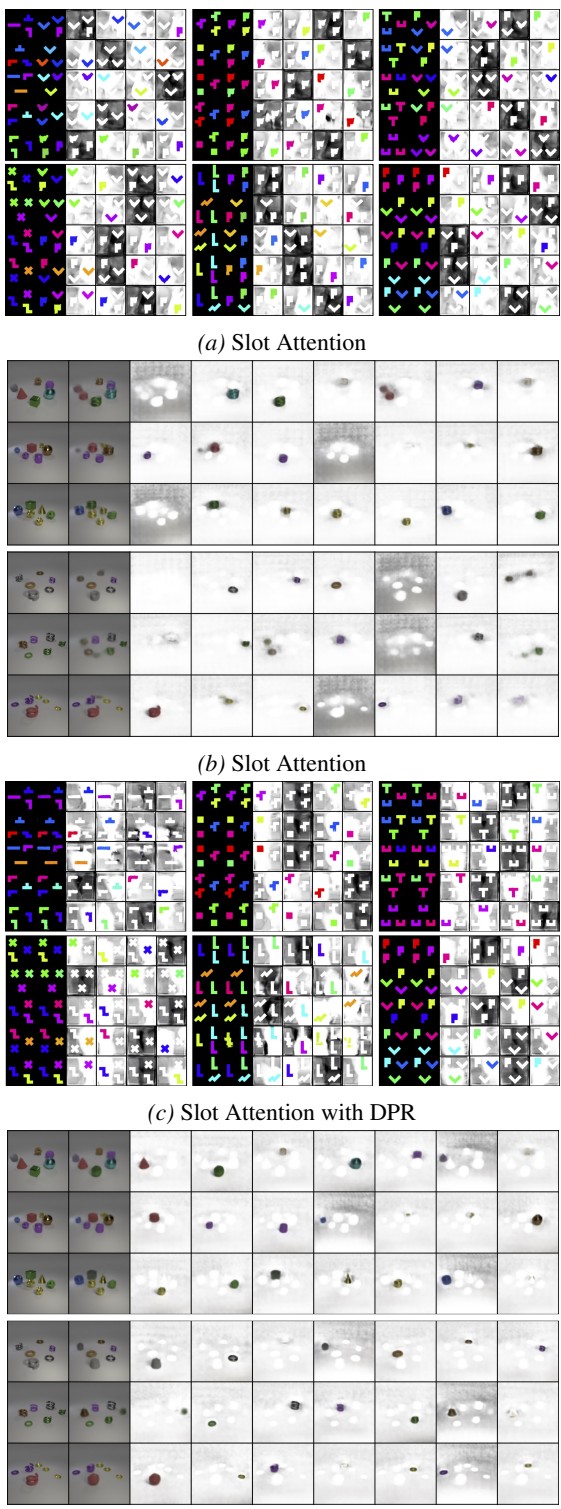

*(a)* Slot Attention

*(b)* Slot Attention

*(c)* Slot Attention with DPR

*(d)* Slot Attention with DPR

*Figure 13.* Reconstruction results of Slot Attention using with (a,b) and without DPR (c,d) on C-Tetrominoes MST, and CLEVR MST. We used the model after continuously training the last task ($\mathcal{T}_5$). Figures are in order of sequential tasks from $\mathcal{T}_0$ to $\mathcal{T}_5$ for Tetrominoes, and we only report results of initial $\mathcal{T}_0$ and last $\mathcal{T}_5$ task for CLEVR.

## A.6. Ablation on Replay Strategies.

We conduct an ablation study on the replay strategy by evaluating Post Replay (PR), which relaxes the *decoder-only* constraint of DPR by updating the full model. Detailed algorithm for PR is provided in Algorithm 2. We compare PR and DPR with standard replay methods described below.

**Replay-based Methods in Continual Learning**   Experience Replay (Lin, 1992) mitigates catastrophic forgetting by jointly training on a mixture of current task data and stored samples from previous tasks within each mini-batch. Subsequent variants (Riemer et al., 2018; Lopez-Paz & Ranzato, 2017b; Chaudhry et al., 2018; Buzzega et al., 2020b; Rolnick et al., 2019; Kumari et al., 2022) have aimed to enhance replay effectiveness by improving sample quality or sampling strategies. However, the question of *when* to incorporate replay samples has received comparatively less attention. Some prior works have explored two-stage strategies (Li et al., 2024; Liu et al., 2023; Wang et al., 2023a; Gupta et al., 2020; Unal et al., 2023; Wang et al., 2024; ji et al., 2024), either by introducing auxiliary modules (Wang et al., 2023a; 2024) or by jointly training with current and replay samples in separate phases (Li et al., 2024; Liu et al., 2023; ji et al., 2024). Notably, (Gupta et al., 2020; Unal et al., 2023) proposed distinct stages of current-task training and jointly training with current and replay samples, while performing balancing to alignment between tasks. We emphasize the importance of replay scheduling, we propose *Post Replay*, which trains on replay samples and a subset of current-task data *after* completing the main training on the current task.

**Comparison with Standard Replay Method.**   For comparison, we follow standard replay methods in continual learning. We apply standard Experience Replay (ER) to Slot Attention. We also evaluate a generative variant motivated by Generative Replay (GR) methods (Shin et al., 2017; Wu et al., 2018; Ayub & Wagner; Zhai et al., 2019; Gao & Liu, 2023), which utilize an auxiliary generative model to produce replay samples. and performs joint training with generated samples.

The results in Table 19 suggest that while DPR provides a minimalist implementation ensuring factor-wise homogeneity, further gains are achievable via PR, provided that this homogeneity remains intact. Notably, both DPR and PR demonstrate performance comparable to ER, even though ER integrates replay samples directly during the novel task training phase.

*Table 19.* Ablation studies to replay strategy and comparison with standard replay methods in continual learning. FG-ARI($_{\pm\text{std.}}$) results on previous tasks $\mathcal{T}_i$ (E$i$) evaluated after training on novel task $\mathcal{T}_j$ (T$j$). SA$^\dagger$ denotes the baseline Slot Attention.

| Model | C-Tetrominoes MST | | | | | |
|---|---|---|---|---|---|---|
| | E0 / T5 | E1 / T5 | E2 / T5 | E3 / T5 | E4 / T5 | E5 / T5 |
| SA$^\dagger$ Joint | $99.84_{\pm.00}$ | $99.84_{\pm.00}$ | $99.81_{\pm.00}$ | $99.84_{\pm.00}$ | $99.80_{\pm.00}$ | $99.84_{\pm.00}$ |
| SA$^\dagger$ | $49.80_{\pm.03}$ | $61.15_{\pm.06}$ | $48.94_{\pm.08}$ | $36.92_{\pm.06}$ | $41.70_{\pm.06}$ | $99.83_{\pm.00}$ |
| SA$^\dagger$ + ER | $99.86_{\pm.00}$ | $99.84_{\pm.00}$ | $\mathbf{99.70}_{\pm.00}$ | $99.51_{\pm.00}$ | $99.31_{\pm.00}$ | $99.30_{\pm.00}$ |
| SA$^\dagger$ + GR | $50.42_{\pm.01}$ | $65.28_{\pm.07}$ | $58.09_{\pm.02}$ | $44.93_{\pm.13}$ | $39.92_{\pm.04}$ | $99.85_{\pm.00}$ |
| SA$^\dagger$ + PR | $\mathbf{99.87}_{\pm.00}$ | $99.64_{\pm.00}$ | $99.49_{\pm.00}$ | $\mathbf{99.68}_{\pm.00}$ | $\mathbf{99.73}_{\pm.00}$ | $99.73_{\pm.00}$ |
| SA$^\dagger$ + DPR | $96.41_{\pm.01}$ | $\mathbf{98.87}_{\pm.01}$ | $98.48_{\pm.00}$ | $98.04_{\pm.01}$ | $97.93_{\pm.01}$ | $99.10_{\pm.01}$ |

### A.7. Architectural Ablations of Factor-wise Homogeneity and Inter-Task Separation

We investigate which architectural components of Slot Attention contribute most significantly to the emergence of factor-wise homogeneity. We focus on the following key components: (1) *Random initialization* of slots (BO-QSA (Jia et al.)), (2) *GRU updates*, and (3) *Residual MLPs*. To evaluate the individual contribution of each, we conduct targeted ablation studies by selectively disabling or modifying these modules. Detailed configurations are provided in Appendix B.4.

**Impact on Inter-task Separation.** As shown in Fig. 14, ablating random initialization (BO-QSA) or the residual MLP yields marginal changes. In contrast, removing the GRU component results in a substantial degradation in inter-task separation across both metrics, leading to inter-task overlaps comparable to those of the baseline SlotMLP. These results are also evident in the visualizations (Fig. 15), which show increased inter-task overlap and reduced separation among slot representations when the GRU is removed.

These results support the findings in Section 5.4, where the GRU serves as the core mechanism for the organization of the latent slot space. It demonstrates object-conditioned updates (slot modes) that are consistent across both factors and tasks. Consequently, we identify the GRU update as the critical architectural component for the emergence of factor-wise homogeneity and inter-task separation.

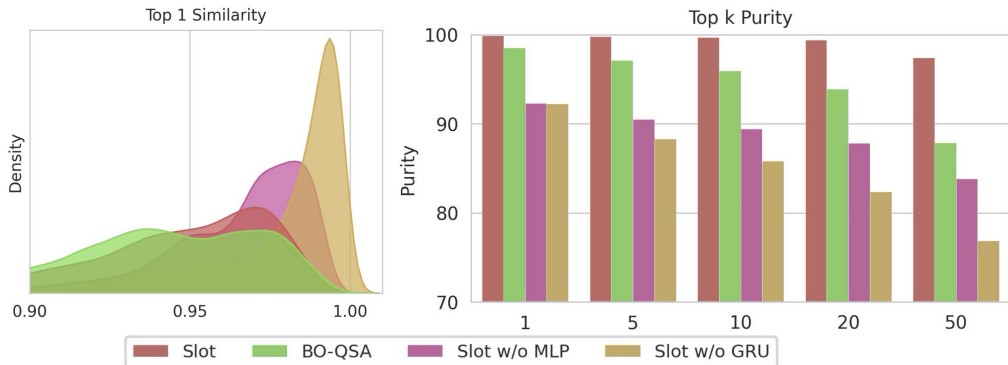

*Figure 14.* Inter-task separation is measured using Density of Top-$k$ Inter-Task Class Similarity per point (left), and Top-$k$ Nearest Neighbor Class Purity (right). Term 'Slot' denotes Slot Attention.

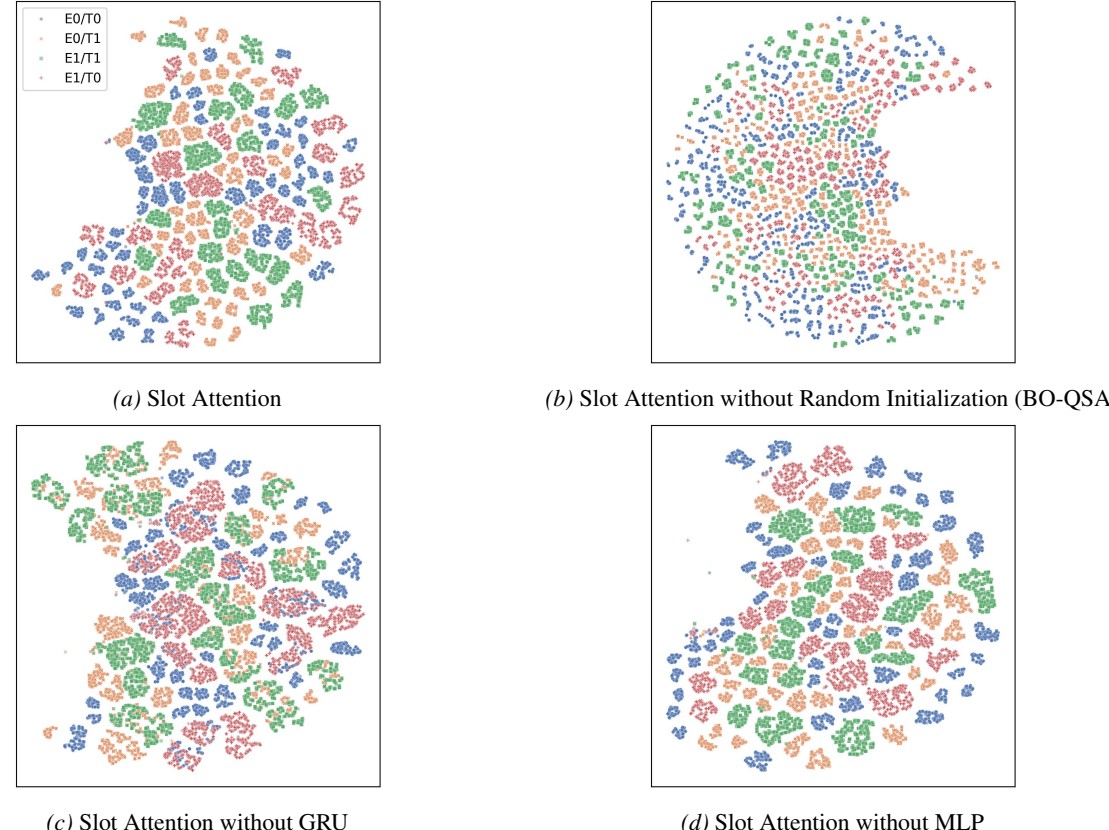

*(a)* Slot Attention

*(b)* Slot Attention without Random Initialization (BO-QSA)

*(c)* Slot Attention without GRU

*(d)* Slot Attention without MLP

*Figure 15.* Slot inter-task separation t-SNE visualization of ablating each modules of Slot Attention .

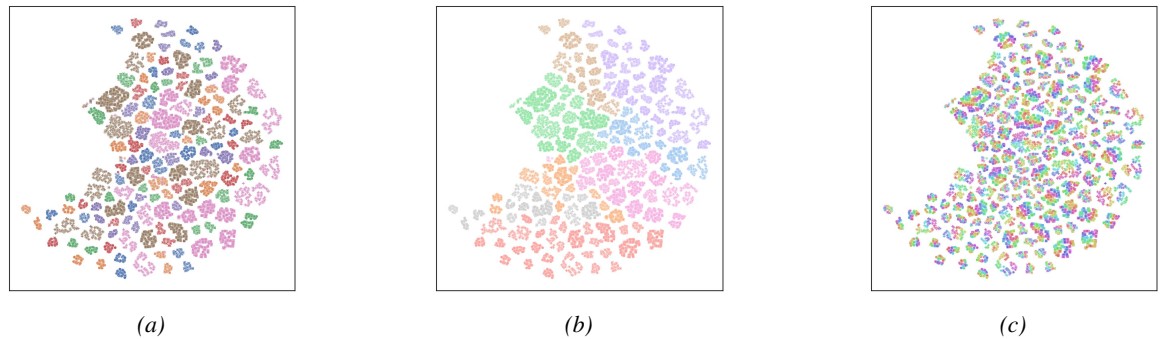

*(a)*

*(b)*

*(c)*

*Figure 16.* t-SNE visualization results of slots from Slot Attention. Each color of the dots represent (a) *shape*, (b) *position*, (c) *color*.

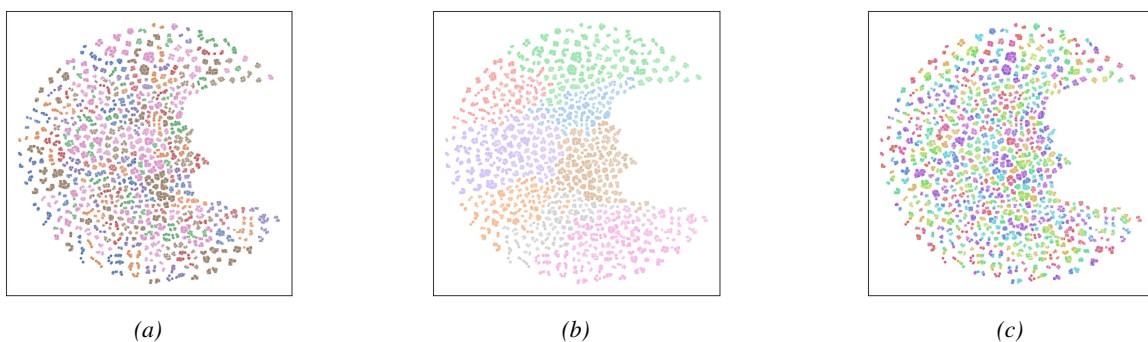

*(a)*      *(b)*      *(c)*

*Figure 17.* t-SNE visualization results of slots from Slot Attention without random initialization (BO-QSA (Jia et al.)). Each color of the dots represent (a) *shape*, (b) *position*, (c) *color*.

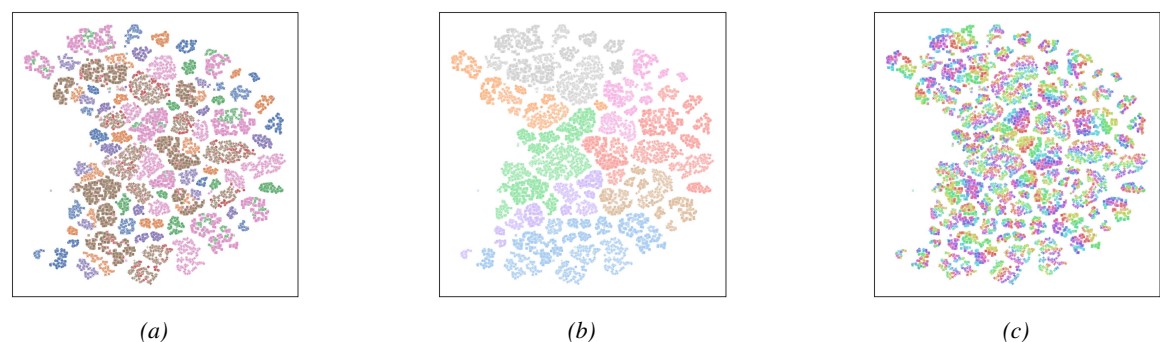

*(a)*      *(b)*      *(c)*

*Figure 18.* t-SNE visualization results of slots from Slot Attention without GRU. Each color of the dots represent (a) *shape*, (b) *position*, (c) *color*.

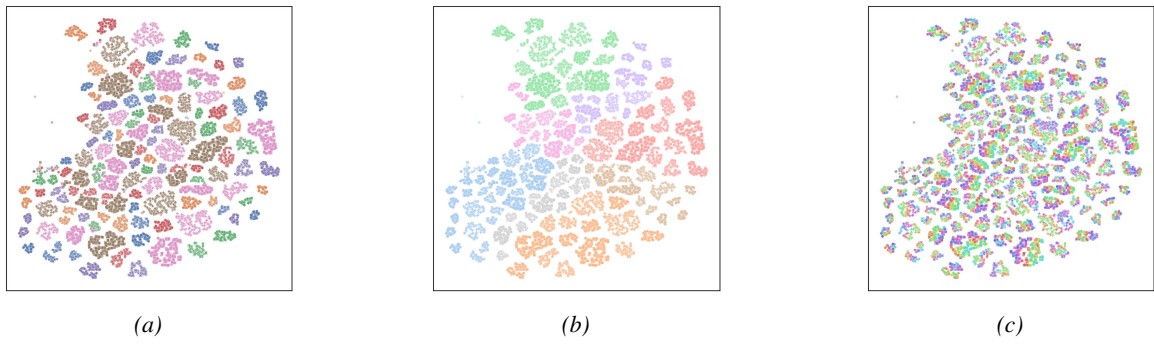

*(a)*      *(b)*      *(c)*

*Figure 19.* t-SNE visualization results of slots from Slot Attention without residual MLP. Each color of the dots represent (a) *shape*, (b) *position*, (c) *color*.

## A.8. Robustness to Continual Learning Regularizer

**Comparison with Standard Continual Regularization.**    In addition, we compare DPR with regularization-based methods, commonly used in class-incremental learning. We evaluate DPR in comparison with Slot Attention combined with Learning without Forgetting (LwF) (Li & Hoiem, 2017) and Elastic Weight Consolidation (EWC) (Kirkpatrick et al., 2017). To assess the compatibility and robustness of DPR with respect to these strategies, we also report results for hybrid models that combine DPR with each regularization method. Detailed configurations for each implementations is described at Section. B.4 (LwF) and Section. B.4 (EWC).

**Results.**    Table 20 presents comparisons and combinations with regularization methods in continual learning. Compared to DPR, LwF and EWC are less effective for reconstruction when applied to Slot Attention, as they continue to suffer performance degradation after training on new tasks. Interestingly, combining them with DPR (SA+EWC+DPR) yields slightly improved performance on MSE metric over using DPR alone, suggesting that DPR is robust to integration with regularization in continual learning. Results for other scenarios are provided in Fig. 20.

*Table 20.* Comparison and integration with regularizer in continual learning. FG-ARI$(_{\pm\text{std.}})$ results on previous tasks $\mathcal{T}_i$ (E$i$) evaluated after training on novel task $\mathcal{T}_j$ (T$j$). SA$^\dagger$ denotes the baseline Slot Attention.

| Model | *C-Tetrominoes MST* | | | | | |
|---|---|---|---|---|---|---|
| | E0 / T5 | E1 / T5 | E2 / T5 | E3 / T5 | E4 / T5 | E5 / T5 |
| SA$^\dagger$ Joint | $99.84_{\pm.00}$ | $99.84_{\pm.00}$ | $99.81_{\pm.00}$ | $99.84_{\pm.00}$ | $99.80_{\pm.00}$ | $99.84_{\pm.00}$ |
| SA$^\dagger$ | $49.80_{\pm.03}$ | $61.15_{\pm.06}$ | $48.94_{\pm.08}$ | $36.92_{\pm.06}$ | $41.70_{\pm.06}$ | $99.83_{\pm.00}$ |
| SA$^\dagger$ + EWC | $47.69_{\pm.18}$ | $52.50_{\pm.15}$ | $52.32_{\pm.09}$ | $32.43_{\pm.17}$ | $39.33_{\pm.17}$ | $84.77_{\pm.05}$ |
| SA$^\dagger$ + LWF | $45.70_{\pm.04}$ | $52.11_{\pm.12}$ | $32.43_{\pm.04}$ | $39.33_{\pm.08}$ | $39.33_{\pm.01}$ | $99.23_{\pm.00}$ |
| SA$^\dagger$ + EWC + DPR | $93.57_{\pm.03}$ | $96.58_{\pm.02}$ | $94.35_{\pm.05}$ | $93.15_{\pm.05}$ | $94.91_{\pm.02}$ | $96.31_{\pm.02}$ |
| SA$^\dagger$ + LWF + DPR | $\mathbf{97.82}_{\pm.01}$ | $98.17_{\pm.00}$ | $98.04_{\pm.00}$ | $95.99_{\pm.01}$ | $93.29_{\pm.01}$ | $93.98_{\pm.01}$ |
| SA$^\dagger$ + DPR | $96.41_{\pm.01}$ | $\mathbf{98.87}_{\pm.01}$ | $\mathbf{98.48}_{\pm.00}$ | $\mathbf{98.04}_{\pm.01}$ | $\mathbf{97.93}_{\pm.01}$ | $\mathbf{99.10}_{\pm.01}$ |

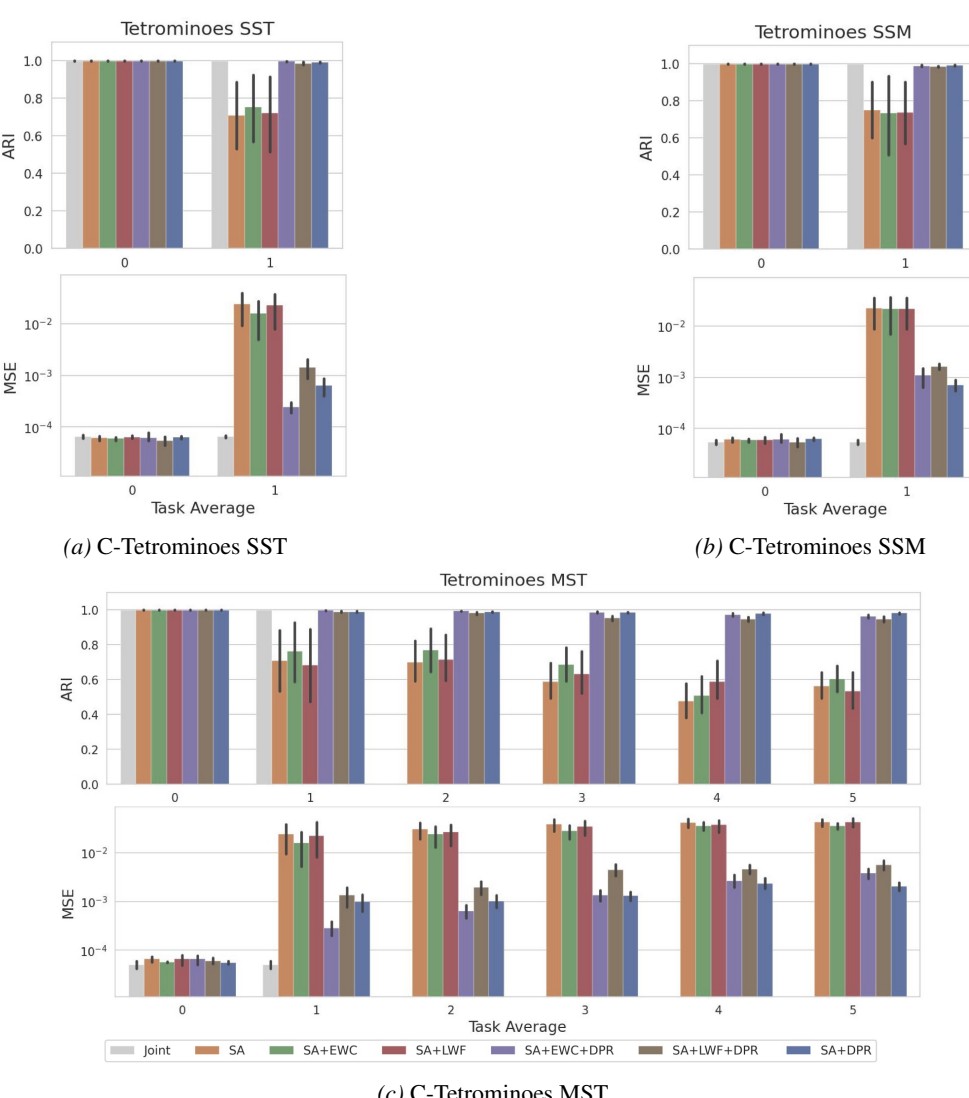

*(a)* C-Tetrominoes SST

*(b)* C-Tetrominoes SSM

*(c)* C-Tetrominoes MST

*Figure 20.* Comparison with continual regularizer-based methods.

## A.9. Analysis on GRU Dynamics and Factor-Wise Homogeneity

In Section 4, we found factor-wise homogeneity property of Slot Attention, where slots consistently preserve identical semantic factor values while differentiating with different factors. This property extends beyond a single task and persists across continual tasks, preserving inter task separation which provides a strong inductive bias for continual learning scenarios, mitigating performance degradation caused by representation overlap.

In this section we further analyze the emergence of factor-wise homogeneity property of Slot Attention and inter task separation in continual settings. From Table 11 we observed the influence of the GRU. While the MLP shows high diversity, its low Frobenius Norm indicates a passive contribution. The GRU, however, combines the highest Frobenius Norm with comparable diversity, suggesting that it serves as the core mechanism for organization of the latent slot space.

The main question we address here is whether the recurrent update dynamics of the GRU are responsible for the stability of factor-wise homogeneity under continual adaptation. To empirically analyze this phenomenon, we leverage recent theoretical studies on time-scale dynamics in gated RNNs, which demonstrate that gating mechanisms effectively create "slow modes"—hidden-state dimensions that evolve slowly and stably, serving as reliable channels for long-term information preservation (Tallec & Ollivier, 2018; Krishnamurthy et al., 2022; Can et al., 2020; Smith et al., 2021; Miller & Hardt). We hypothesize that these slow modes are utilized by Slot Attention to stabilize factor-specific information.

We employ the GRU update gate $z_t$ as an empirical measure of dimensional stability (a 'slow mode rank') and compute the Spearman's rank correlation across slots to confirm that slots sharing identical semantic factors rely on the same highly stable dimensions. Our analysis confirms that the GRU effectively performs an identical-mapping operation conditioned on each object's intrinsic variability, stabilizing factor-specific slow modes. Furthermore, these stability characteristics are robustly maintained under continual learning conditions, providing structural foundation for the robust task-level separation observed in Slot Attention. This slow-mode mechanism successfully explains both the factor-wise homogeneity and the task separation observed in Slot Attention under continual learning.

**Time-scale dynamics in gated RNNs**    A substantial body of work has shown that gated recurrent architectures inherently produce multi–time-scale dynamics in their hidden states. Theoretical analyses demonstrate that gating mechanisms effectively learn local time-steps, causing certain hidden-state dimensions to evolve slowly and thus act as stable carriers of long-range information (Tallec & Ollivier, 2018). Subsequent studies provide direct evidence that gated RNNs generate marginally stable directions that persist across extended horizons, forming slow modes that dominate the network's long-term behavior (Krishnamurthy et al., 2022). Complementary dynamical-systems perspectives show that gates induce eigen-directions with eigenvalues clustered near one, enabling long-timescale stability and supporting continuous attractor structures within the hidden-state space (Can et al., 2020). (Smith et al., 2021) propose a reverse-engineering framework based on Jacobian switching linear dynamical systems, showing that trained RNNs can be locally approximated by low-dimensional linear dynamics controlled by a small set of dominant eigen-directions. Importantly, many of these directions evolve slowly, revealing that long-horizon behavior in RNNs is governed by a compact set of stable dynamical modes. In parallel, (Miller & Hardt) analyze stability properties of recurrent models and demonstrate that trained RNNs often operate near nearly-identity transformations. Such near-identity dynamics naturally give rise to slowly changing hidden-state dimensions, providing an inherent mechanism for maintaining semantically meaningful information over time. Taken together, these findings establish that gated RNNs reliably form persistent, slowly evolving subspaces—a phenomenon directly aligned with our use of GRU update-gate behavior to characterize slow modes.

**Empirical Analysis for GRU Dynamics of Slot Attention**    Because our ablation study suggested that removing the GRU disrupts this homogeneity, we further investigate whether the recurrent updates are responsible for its stability. We adopt the theoretical perspective that the gating mechanisms in RNNs, such as the GRU, effectively generate "slow modes"—hidden-state dimensions whose update gate ($z_t$) remains consistently small and thus changes minimally across iterations. These slow modes act as stable structural channels for preserving key semantic information over long horizons. In this view, the theory establishes that a "slow mode" corresponds to a hidden-state dimension whose update gate ($z_t$) remains consistently small, as denoted by the final update (Equation 5):

$$h_t = (1 - z_t) \odot h_{t-1} + z_t \odot \tilde{h}_t. \tag{5}$$

We hypothesize that this stability is the theoretical basis for the emergence of stable, factor-specific representations. We leverage this stability property by using the GRU update gate $z_t$ as an empirical measure of dimensional stability (a "slow

mode rank"). We then apply this stability measure to analyze factor-wise homogeneity by correlating these 'slow mode ranks' across slots to confirm that slots sharing identical semantic factors rely on the same highly stable (slow mode) dimensions.

To assess consistency between semantic factors, we analyze the slot representation of the $i$-th object, denoted as $o^i_{s,c,p}$ as in Fig. 3 (where $s$ is shape, $c$ is color, and $p$ is position). We collect GRU update-gate activations $z^i_{t,d}$ (Equation 5) for each slot dimension $d \in 1, ..., D$ and compute a slow mode rank vector $r^i \in \mathbb{R}^D$. We then compute the Spearman's rank correlation (Equation 6) to analyze whether the slow modes of each dimension show consistency of the ordering (rank) of these stable dimensions across slots with respect to factors:

$$\rho = \text{Spearman}\left(r^i, r^j\right) \tag{6}$$

**GRU Slow Mode Consistency Across Semantic Factors**   We first seek to validate if the stability of the GRU's slow mode dimensions directly corresponds to semantic factors and remains highly consistent across slots sharing identical factors. This test confirms the meaningfulness of our theoretical approach in explaining the observed structural robustness. For object in intra-tasks where evaluation is done on models that is trained on identical task (e.g. E0/T0, E1/T1), we compute correlations between (1) object pairs with identical factors and (2) object pairs with different factors.

*Table 21.* Spearman's rank correlation ($\rho$) of GRU slow mode ranks for objects within the same task. The high correlation for identical factors confirms the consistency of factor-specific stable dimensions.

| Evaluation Setting $(E_i/T_j)$ | Objects with identical factors $(\rho)$ | Objects with different factors $(\rho)$ |
|---|---|---|
| E0/T0 | 0.9921 | 0.7931 |
| E1/T1 | 0.9920 | 0.7958 |

As shown in Table 21, objects sharing identical factors exhibit a nearly perfect correlation ($\rho \approx 0.99$), showing that slow-mode GRU dimensions are highly consistent across slots with the same semantics. Conversely, although correlations for different-factor pairs are numerically substantial (e.g., $\rho \approx 0.79$), the consistent and significant gap indicates the presence of distinctive factor-specific slow-mode dimensions. These findings directly align with the geometric observation in Fig. 3, confirming that the GRU effectively performs an identical-mapping operation conditioned on each object's intrinsic variability, stabilizing slots that share the same semantics along these consistent slow-mode dimensions.

**Preservation of Slow Mode Under Continual Learning**   Finally, we examine whether the observed structural robustness persists under continual settings, which is essential for maintaining task separation. The primary purpose of this analysis is to test whether the GRU slow modes maintain the representation quality (intra-task retention of previous task) and remain consistent during continual settings (inter-task separation between previous task and novel task). We compute correlations for (1) samples from the previous task (intra-task retention) and (2) inter-samples across previous and novel tasks (inter-task separation). We compare these results with the earlier correlations (Table 21), treating them as the baseline under standard (non-continual) conditions.

*Table 22.* Spearman's rank correlation ($\rho$) of GRU slow mode ranks under the continual learning setting. Note that *shape* factors for all sequential tasks are disjoint, leading to no objects with identical factors for inter-task comparisons (bottom row).

| Evaluation Setting $(E_i/T_j)$ | Objects with identical factors $(\rho)$ | Objects with different factors $(\rho)$ |
|---|---|---|
| E0/T1 | 0.9935 | 0.8054 |
| E0/T1 vs E1/T1 | - | 0.7929 |

Results in Table 22 strongly support the preservation of factor-wise homogeneity through the GRU dynamics: Intra-Task Retention: The intra-task correlation for the previous task (E0/T1) remains high and consistent with the baseline, indicating that the GRU performs an identical-mapping operation even for previously seen samples. Inter-Task Separation: Crucially, the inter-task correlation (E0/T1 VS E1/T1) closely matches the correlation observed for objects with different semantic factors under the standard condition. This demonstrates that both tasks retain their distinctive factor-specific slow-mode dimensions, confirming that structural factor-wise separation is robustly maintained under continual settings. These

consistency demonstrates that the GRU dynamics provide a robust mechanism for task-level separation by stabilizing factor-specific slow modes, which is the strong inductive bias for continual object-centric learning.

**Summary: GRU Dynamics and Structural Robustness**   Motivated by the degradation observed in our ablation study, this section aimed to analyze the emergence of factor-wise homogeneity and task-level separation in continual learning through the lens of GRU slow modes. We applied the slow mode rank, derived from the GRU update gate, as an empirical measure of dimensional stability. Our analysis demonstrated two key findings: (1) Slots sharing identical semantic factors exhibit a nearly perfect correlation ($\rho \approx 0.99$) in their slow mode ranks and distinctive gap between correlation among slots with different factors (Table 21), confirming that the GRU stabilizes shared semantic factors along consistent dimensions. (2) Under the continual learning setting, the inter-task correlation between old and new tasks remains low (Table 22), robustly maintaining the task separation observed in the single-task baseline.

Our analysis shows that the GRU's marginally stable slow modes effectively perform an identical-mapping operation conditioned on each object's intrinsic variability, stabilizing slots that share the same semantic factors. This slow-mode mechanism successfully explains both the robustness of factor-wise homogeneity and the natural task separation observed in Slot Attention under continual learning. This consistency demonstrates that the GRU dynamics provide a robust mechanism for task-level separation by stabilizing factor-specific slow modes, which is the necessary structural foundation for effective continual object-centric learning.

## A.10. Performance on More Challenging Settings

In this section, we aim to evaluate whether the proposed *factor-wise homogeneity* and DPR extends to challenging real-world environments. We first describe the dataset configurations used for continual learning benchmarks (COCO and PASCAL), then present the baseline setting with DINOSAUR as a representative slot-based model for complex images. Finally, we report and analyze the experimental results to demonstrate that our inductive bias remains effective beyond synthetic domains.

**Datasets and evaluation settings**   We extend our evaluation of *factor-wise homogeneity* to more challenging environments to assess whether this property persists beyond simplified synthetic benchmarks. To systematically increase task complexity, we consider two complementary axes: (i) real-world datasets, which introduce diverse and uncontrolled visual variations, and (ii) complex synthetic datasets, which exhibit high variability in texture and background.

For the real-world setting, we adopt two widely used real-world benchmarks in OCL (Seitzer et al.; Didolkar et al., 2025; Kakogeorgiou et al., 2024): MS COCO 2017, which contains multiple objects with diverse labels, and PASCAL VOC 2012, which primarily consists of single labeled objects per image. Since COCO and PASCAL are originally defined as single-task datasets, we follow prior works (Shmelkov et al., 2017; Michieli & Zanuttigh, 2019; Cermelli et al., 2020) and split each dataset into $|\mathcal{T}|$ disjoint tasks according to object categories. We consider two continual learning scenarios: (i) Single-Step addition of Multiple classes (SSM), and (ii) Multi-Step addition of Two classes (MST). We set $|\mathcal{T}| = 2$ for SSM and $|\mathcal{T}| = 4$ for MST, ensuring that each task includes the same number of object categories, divided alphabetically. Following the disjoint settings of prior works, we retain only images whose object labels do not overlap across tasks. Table 23 summarizes the dataset configurations used in our evaluation.

For the complex synthetic setting, we construct a hybrid scenario that combines CLEVRTex (Karazija et al.), which introduces diverse textures and realistic backgrounds, with Continual-CLEVR. Specifically, we train the model on the initial task $\mathcal{T}_0$ of Continual-CLEVR and subsequently introduce CLEVRTex as the second task. To ensure a fair comparison, we sample the same number of images from CLEVRTex as in $\mathcal{T}_0$, restricting to scenes with at most six objects. This setup allows us to evaluate whether *factor-wise homogeneity* remains effective when transferring from a relatively simple synthetic environment to one with significantly richer visual complexity.

**Baseline Setting**   To evaluate on challenging real-world images, we adopt COCO (Lin et al., 2014) and PASCAL VOC (Everingham et al., 2010), two widely used benchmarks in OCL. Since Slot Attention does not scale well to complex synthetic or real-world data (Seitzer et al.), we employ DINOSAUR (Seitzer et al.) as our baseline — a slot-based model integrated with pre-trained DINO (Caron et al., 2021) that surpasses vanilla Slot Attention and its variants on real-world images. DINOSAUR follows an autoencoder framework similar to vanilla Slot Attention, but instead of reconstructing the input image, it reconstructs latent patch features $h_i \in \mathbb{R}^{N \times D}$ from a target encoder (e.g., pre-trained DINO). This enables the slots to capture more high-level semantics, leading to significant improvements on complex images.

*Table 23.* Dataset configuration for COCO and PASCAL on continual setting

| Dataset | Scenario | $|\mathcal{T}|$ | Objects per Task | Total Objects |
|---------|----------|-----|------------------|---------------|
| COCO | SSM | 2 | 40 | 80 |
|      | MST | 4 | 20 | 80 |
| PASCAL | SSM | 2 | 10 | 20 |
|        | MST | 4 | 5 | 20 |

*Table 24.* Reproduced performance of DINOSAUR on real-world COCO and PASCAL VOC dataset (5 runs, mean$_{\pm\text{std}}$).

| Dataset | Model | FG-ARI | mBO$^c$ | mBO$^i$ |
|---------|-------|--------|---------|---------|
| COCO | DINOSAUR | $32.37_{\pm 0.8}$ | $42.94_{\pm 0.7}$ | $31.25_{\pm 0.7}$ |
| PASCAL | DINOSAUR | $23.63_{\pm 0.7}$ | $52.96._{\pm 0.8}$ | $46.29_{\pm 0.7}$ |

As a baseline configuration, we use ImageNet (Deng et al., 2009) pre-trained DINO-Base/16 as both the feature extractor and target encoder, following the setup in (Seitzer et al.). We train the network with Adam (Kingma & Ba, 2014) using a learning rate of $4 \times 10^{-4}$, a warm-up phase, and an exponential decay schedule. We adopt a global batch size of 64 and set 7 slots for COCO and 6 slots for PASCAL. We adopt a global batch size of 64 and set 7 slots for COCO and 6 slots for PASCAL. The reproduced performance of DINOSAUR is reported in Table 24, and we use the same configurations for all following experiments using DINOSAUR. Results are evaluated using Foreground Adjusted Rand Index and mean Best Overlap (Pont-Tuset et al., 2016), averaged over 5 independent runs.

Since DINOSAUR reconstructs latent patch features from the target encoder rather than pixel-level inputs as in vanilla Slot Attention, it does not directly output image reconstructions. To enable decoder fine-tuning in the DPR phase, we store a subset of randomly sampled input images as a replay buffer. We use the same buffer size as in the synthetic benchmarks.

For CLEVRTex, we follow the same configuration as in CLEVR (Table 31) and adopt the hyper-parameters in Table 28, ensuring consistency across synthetic experiments. We report results of ARI and MSE averaged over 5 independent runs.

**Factor-wise Homogeneity in real-world Image** Table 5 and Table 25 present results on continual learning with real-world datasets, consisting of four and two sequential tasks, respectively. Our method achieves consistent performance improvements across both datasets, indicating that our findings are not limited to simplified synthetic benchmarks but also hold in real-world scenarios. This further supports our main claim that *factor-wise homogeneity* serves as an effective inductive bias for continual object-centric learning.

In DINOSAUR, both the feature extractor and target decoder leverage DINO pre-trained on large-scale external resources (ImageNet), which substantially enhance baseline continual learning performance by implicitly providing essential information for new tasks. While this strong prior can obscure the benefits of inductive biases such as factor-wise homogeneity and DPR, we emphasize that DPR yields significant and consistent gains in settings without such external information, and even with DINOSAUR the improvements remain consistent. Moreover, as the number of continual learning steps increases, DINOSAUR's performance on earlier tasks degrades, widening the gap across tasks and revealing its limitations in retaining prior representations (compared to Table 5). In these cases, our method provides increasingly larger gains compared to the results in Table 25. These findings confirm that *factor-wise homogeneity* and DPR act as effective inductive biases for continual object-centric learning, even under strong pre-trained representations such as DINO.

We also visualize the slot distributions of DINOSAUR trained on continual COCO ($T = 2$), using t-SNE as described in Section 4.2. Representations are collected across different tasks $\mathcal{T}_t$. We label them as *task pairs* (E$i$/T$j$), representing evaluation on task $\mathcal{T}_i$ using a model continuously trained from the initial task up to task $\mathcal{T}_j$. Fig. 21 shows slots from different tasks in real-world images preserve *factor-wise homogeneity* and remain well separated across tasks, consistent with result using C-Tetrominoes (Fig. 1a). We observe that slots remain distinct between (i) upcoming tasks with unseen objects (e.g., red vs. green or purple) and (ii) previous tasks without overlaps in continual training (e.g., blue vs. purple). Moreover, slots corresponding to the same evaluation task but from different training phases also remain separated.

Fig. 22 presents qualitative mask-prediction results on the COCO dataset. From these examples, we observe three consistent patterns. (1) In the first three (column-wise) images, DINOSAUR fails to mask objects that appeared in previous tasks but are absent in the current task (e.g., "bicycle", "banana", "sandwich"), whereas DINOSAUR with DPR successfully

*Table 25.* Effectiveness of DPR on real-world images. We evaluate on $\mathcal{T}_0$ (E0) after training on $\mathcal{T}_0$ (T0, Initial) and after continuous training up to $\mathcal{T}_1$ (T1, Final) across $|\mathcal{T}| = 2$ tasks (mean$_{\pm\text{std}}$). DPR$^{\dagger\dagger}$ denotes DPR with DINOSAUR.

| Dataset | Model | E0 / T0 | | E0 / T1 | |
| --- | --- | --- | --- | --- | --- |
| | | FG-ARI | mBO$^c$/mBO$^i$ | ARI | mBO$^c$/mBO$^i$ |
| COCO | DINOSAUR | $22.84_{\pm0.8}$ | $42.93_{\pm0.6}$ / $31.25_{\pm0.7}$ | $21.02_{\pm0.1}$ | $40.63_{\pm0.4}$ / $29.04_{\pm0.4}$ |
| | DPR$^{\dagger\dagger}$ | | | $\mathbf{22.51}_{\pm0.5}$ | $\mathbf{41.74}_{\pm0.2}$ / $\mathbf{30.57}_{\pm0.5}$ |
| PASCAL | DINOSAUR | $16.61_{\pm0.6}$ | $53.47_{\pm0.5}$ / $47.74_{\pm0.6}$ | $15.45_{\pm1.2}$ | $51.79_{\pm1.6}$ / $46.61_{\pm1.2}$ |
| | DPR$^{\dagger\dagger}$ | | | $\mathbf{16.58}_{\pm0.9}$ | $\mathbf{52.70}_{\pm0.7}$ / $\mathbf{47.37}_{\pm0.5}$ |

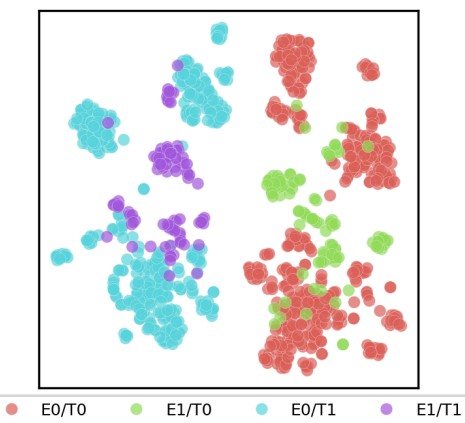

*Figure 21.* Inter-task separation visualization of Slots from DINOSAUR via t-SNE. We use COCO dataset split into $|\mathcal{T}| = 2$ tasks. Each dot denotes a slot representation from different environments, where evaluation tasks (E$i$) are obtained after continuous training up to task (T$j$).

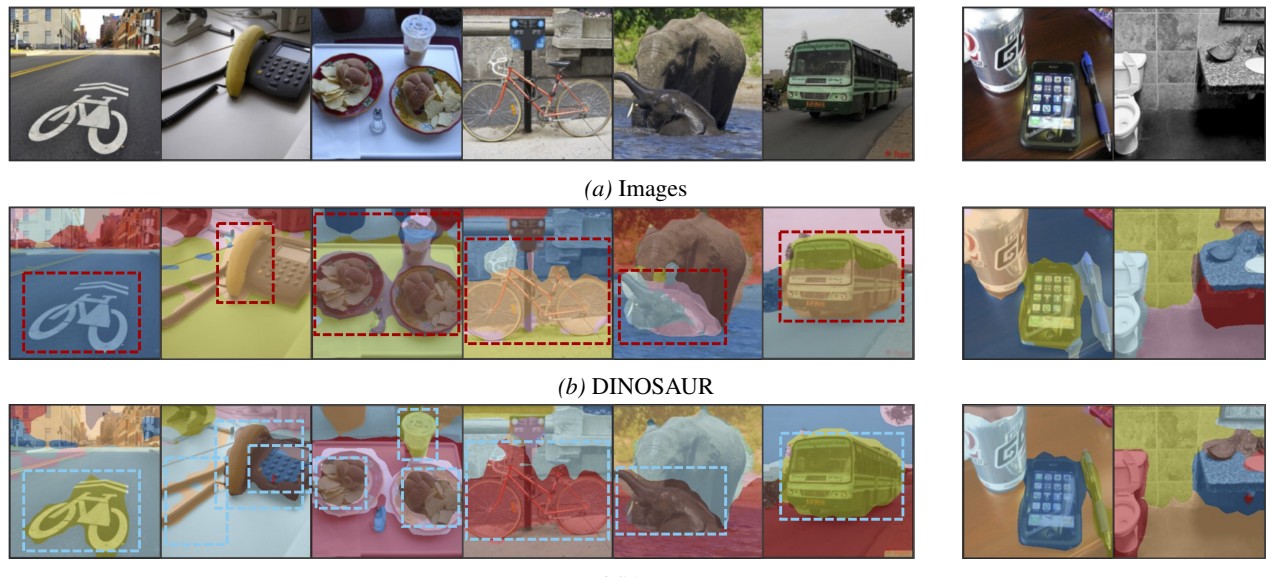

*(a)* Images

*(b)* DINOSAUR

*(c)* DINOSAUR + DPR

*Figure 22.* Example results on COCO dataset with $|\mathcal{T}| = 4$ continual tasks. First six images belong to previous tasks ($\mathcal{T}_{0\sim2}$) after training in novel task ($\mathcal{T}_3$), and last two images are from evaluation set of novel task ($\mathcal{T}_3$). Red bounding boxes highlights the failed points of DINOSAUR and blue bounding boxes highlights modifications after DPR. (1) DINOSAUR fails to mask objects that appeared in previous tasks (e.g., a "bicycle", "banana", "sandwich"), while DPR successfully recovers and masks these objects. (2) Also DPR improves to produce a coherent single-slot mask (e.g., a "bicycle", "elephant", "bus") compared to DINOSAUR. (3) Finally, DPR does not degrade mask quality of novel task (e.g., last two images).

*Table 26.* Performance on transferring environments with CLEVR ($\mathcal{T}_0$) and CLEVR-Tex ($\mathcal{T}_1$). We report FG-ARI$_{\pm\text{std.}}$ and MSE$_{\pm\text{std.}}$ results.

| Model | E0 / T0 | | E0 / T1 | |
|---|---|---|---|---|
| | FG-ARI | MSE | FG-ARI | MSE |
| Slot Attention | $96.51_{\pm.01}$ | $0.0004_{\pm.00}$ | $68.68_{\pm.06}$ | $0.004_{\pm.01}$ |
| DPR$^\dagger$ | | | $\mathbf{82.75}_{\pm.02}$ | $\mathbf{0.001}_{\pm.00}$ |

recovers and masks these objects. In particular, DINOSAUR does not separately mask the "bicycle" or "banana", while DPR clearly separates them from surrounding regions (e.g., separating the "bicycle" from the road and the "banana" from the telephone and telephone line). We further observe cases where DINOSAUR produces integrated mask of "sandwich" while DPR masked both "sandwich" and "cup" of previous task. (2) In the next three images, DPR yields cleaner object-wise segmentation. For instance, DINOSAUR over-segments the baby "elephant" into multiple slots, whereas DPR produces a coherent single-slot mask. Similar improvements are also observed for the "bicycle" and "bus". (3) In the final two images, which belong to the novel task, DPR does not degrade mask quality; both DINOSAUR and DINOSAUR with DPR produce effective masks for unseen objects.

These results further confirm that *factor-wise homogeneity* helps preserve slot representations against task interference, not only in synthetic images but also in complex real-world data.

**Factor-wise Homogeneity in Complex Synthetic Images**   Table 26 presents the results for $\mathcal{T}_0$ (CLEVR) after initial training on $\mathcal{T}_0$ followed by continual training on $\mathcal{T}_1$ (CLEVRTex). Without DPR, Slot Attention exhibits degradation on the evaluation of the previous task ($\mathcal{T}_0$) when trained continuously on the more complex dataset. In contrast, incorporating DPR substantially improves performance, indicating that *factor-wise homogeneity* together with DPR provides a stable and robust inductive bias in complex synthetic environments, extending beyond simple synthetic settings.

# B. Appendix: Implementations and Training Configurations

## B.1. Algorithms

We provide pseudo-code implementations for three replay strategies used in our experiments: (1) Decoder-only Post Replay, (2) Post Replay. Each method differs in which components are updated and when replay is applied, as summarized in Alg. 1, 2.

---

**Algorithm 1** Decoder-only Post Replay for Slot Attention

---

1: **Input:** Encoder $\mathbf{f}_\theta$, Slot Attention $\mathbf{h}_\theta$, Decoder $\mathbf{g}_\theta$, loss $\mathcal{L}$, learning rate $\eta$, replay buffer $\mathcal{R}$
2: **for** $t = 1$ **to** $T$ **do**
3:      Train $\mathbf{f}_\theta$, $\mathbf{h}_\theta$, $\mathbf{g}_\theta$ on $\mathcal{T}_t$                                          *(Main task training phase)*
4:      Sample subset of current task $\mathcal{T}_t^* \subseteq \mathcal{T}_t$
5:      $\mathcal{R}_t^* \leftarrow \mathcal{R} \cup \mathcal{T}_t^*$                                        *(Replay + current task subset)*
6:      Freeze $\mathbf{f}_\theta$, $\mathbf{h}_\theta$
7:      **for** epoch $= 1$ **to** $N_{\text{replay}}$ **do**
8:          Sample mini-batch $B$ from $\mathcal{R}_t^*$                                *(Post replay phase)*
9:          $\mathbf{g}_\theta \leftarrow \mathbf{g}_\theta - \eta \cdot \nabla_{\mathbf{g}_\theta} \mathcal{L}(B; \mathbf{f}_\theta, \mathbf{h}_\theta, \mathbf{g}_\theta)$
10:     **end for**
11:     Store subset of generated samples in buffer $\mathcal{R}_t \subseteq \hat{X}_t$
12:     $\mathcal{R} \leftarrow \mathcal{R} \cup \mathcal{R}_t$
13: **end for**=0

---

**Algorithm 2** Post Replay for Slot Attention

---

1: **Input:** Encoder $\mathbf{f}_\theta$, Slot Attention $\mathbf{h}_\theta$, Decoder $\mathbf{g}_\theta$, loss $\mathcal{L}$, learning rate $\eta$, replay buffer $\mathcal{R}$
2: **for** $t = 1$ **to** $T$ **do**
3:      Train $\mathbf{f}_\theta$, $\mathbf{h}_\theta$, $\mathbf{g}_\theta$ on $\mathcal{T}_t$                                          *(Main task training phase)*
4:      Sample subset of current task $\mathcal{T}_t^* \subseteq \mathcal{T}_t$
5:      $\mathcal{R}_t^* \leftarrow \mathcal{R} \cup \mathcal{T}_t^*$                                        *(Replay + current task subset)*
6:      **for** epoch $= 1$ **to** $N_{\text{replay}}$ **do**
7:          Sample mini-batch $B$ from $\mathcal{R}_t^*$                                *(Post replay phase)*
8:          $\{\mathbf{f}_\theta^*, \mathbf{h}_\theta^*, \mathbf{g}_\theta^*\} \leftarrow \{\mathbf{f}_\theta, \mathbf{h}_\theta, \mathbf{g}_\theta\} - \eta \cdot \nabla_{\mathbf{g}_\theta} \mathcal{L}(B; \mathbf{f}_\theta, \mathbf{h}_\theta, \mathbf{g}_\theta)$
9:      **end for**
10:     Store subset of generated samples in buffer $\mathcal{R}_t \subseteq \hat{X}_t$
11:     $\mathcal{R} \leftarrow \mathcal{R} \cup \mathcal{R}_t$
12: **end for**=0

---

## B.2. Computation Costs of DPR

In this section, we discuss computation overheads due to DPR. We provide an we provide an evaluation using the COCO dataset with Dinosaur model. Critically, DPR does not perform replay during the training phase; it only performs replay to the decoder once training is complete. Let $N_{train}$ and $N_{replay}$ denote the number of training and replay samples where $N_{train} \gg N_{replay}$. We use $E_{train}$ (=200) and $E_{post}$ (=30) to denote the number of epochs for training each task and the post-replay phase. For Dinosaur, which keeps the ViT encoder frozen, there are 41M trainable parameters, whereas the DPR phase updates only 37M parameters since it is restricted to the decoder.

Since $N_{train} \gg N_{replay}$ and the replay phase involves significantly fewer epochs ($E_{train} \gg E_{post}$) while updating only a subset of the model, DPR demonstrates superior efficiency compared to standard replay methods that require replay throughout the training process.

## B.3. Training Implementations

**Hyper-parameters** We provide detailed training configurations used in our experiments. Following prior works (Locatello et al., 2020; Dittadi et al., 2021), we adopt their implementation settings with modifications to accommodate our C-OCL

*Table 27.* Comparisons of computation costs. Note that $^*$ are computed by $(T_{train} + T_{repaly})/(E_{train} + E_{post})$

| Method | Peak VRAM (Mb) | Time (sec) / Epoch | Complexity |
|---|---|---|---|
| Dinosaur | 2096 | 22.23 | $E_{train} * N_{train}$ |
| Dinosaur | 2918 | 37.95 | $E_{train} * (N_{train} + N_{replay})$ |
| Dinosaur | 2096 | $21.17^*$ | $E_{train} * N_{train} + E_{post} * N_{replay}$ |

benchmarks. Table 28 summarizes the default hyperparameters used across all experiments. Note that for C-CLEVR, which uses higher-resolution images ($128 \times 128$), we set the batch size to 32.

*Table 28.* Default hyper-parameters for training.

| Hyperparameter | Default ($\mathcal{T}_0$) | ($\mathcal{T}_{t>0}$) | Post Replay at $t$ | Joint Replay at $t$ |
|---|---|---|---|---|
| Optimizer | Adam (Kingma & Ba, 2014) | - | - | - |
| Learning rate | $4 \times 10^{-4}$ | - | - | - |
| Warm-up steps | 2% of total steps | - | - | - |
| Batch size | 64 | - | - | 32+32 |
| Resolution | ($64 \times 64$) | - | - | - |
| Training epochs | 200 | 100 | 50 | 100 |
| Train | 25000 | 10000 | $2000 \times t$ | $\mathcal{T}_t + 2000 \times (t-1)$ |
| Evaluation | 5000 | 5000 | None | None |
| Buffer size | None. | - | 2000 per $t$ | 2000 per $t$ |

**Grid search on hyper-parameters for DPR**   We provide a comparison of implementation variants of Decoder-Only Post Replay, focusing on two key elements: (1) replay buffer size and (2) number of replay epochs. Fig. 23 presents the results for each variation.

In terms of ARI, a buffer size of 2,000 provides strong performance, while increasing the size beyond 2,000 does not yield further significant improvement. In contrast, reducing the buffer size to 100 leads to noticeable performance degradation. For MSE, the best result is achieved with a buffer size of 5,000, though 2,000 achieves comparable performance. Table 29 confirms that this trend remains consistent in longer sequences (T=6).

For replay epoch ablation, we compare against the baseline of 50 replay epochs, which matches the number of epochs used in task training. Variants with fewer epochs show slight degradation in both ARI and MSE. Based on these findings, we use buffer size of 2,000 and replay epochs of 50 for our DPR.

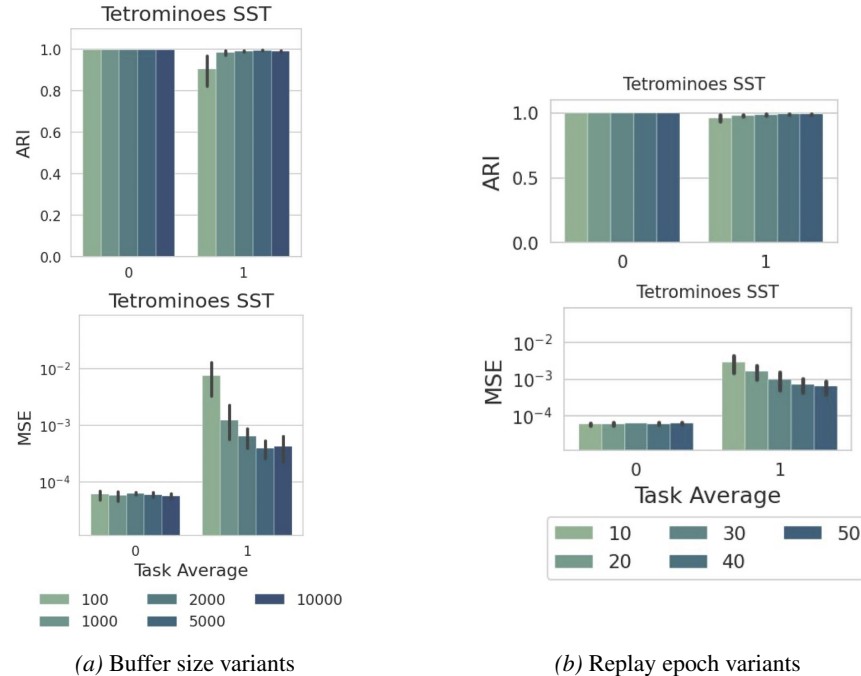

(a) Buffer size variants             (b) Replay epoch variants

*Figure 23.* Grid search on (1) replay buffer size, (2) replay epoch.

*Table 29.* Performance across varying buffer sizes on longer sequential taks.

| Buffer Size | E0 / T5 (FG-ARI) |
|---|---|
| SA | $49.80_{\pm.03}$ |
| SA + DPR 1000 | $93.83_{\pm.04}$ |
| SA + DPR 2000 | $96.41_{\pm.01}$ |
| SA + DPR 5000 | $97.52_{\pm.00}$ |
| SA + DPR 10000 | $98.69_{\pm.01}$ |

## B.4. Model Implementations

**Implementation Slot Attention** We follow the Slot Attention implementation from (Locatello et al., 2020; Dittadi et al., 2021). For the C-Tetrominoes dataset, which has a resolution of $64 \times 64$, we use the architecture described in Table 30. For the C-CLEVR dataset (Table 31, we apply several modifications to handle the larger image resolution of $128 \times 128$. Specifically, we use convolutional channels of size 64 and broadcast the slot representations to a spatial resolution of $8 \times 8$ instead of matching the full input resolution, when feeding them into the broadcast decoder.

*Table 30.* Slot Attention architecture for Tetrominoes ($64\times64$) resolution.

| Module | Type / Operation | Details |
|---|---|---|
| Encoder | Conv $5 \times 5$ | 32 channels, stride 1, padding 2, ReLU |
| | Conv $5 \times 5$ | 32 channels, stride 1, padding 2, ReLU |
| | Conv $5 \times 5$ | 32 channels, stride 1, padding 2, ReLU |
| | Conv $5 \times 5$ | 32 channels, stride 1, padding 2, ReLU |
| | Positional Embedding | Added to conv output |
| | GroupNorm + Conv $1 \times 1$ | 32 channels, ReLU, Flatten |
| Slot Attention | Slot Initialization | Learnable $\mu, \sigma$ |
| | Slot | $D_{slots} = 64$ |
| | GRU Update | GRUCell(dim=64) per slot |
| | MLP | 2-layer FFN (hidden dim=128), ReLU |
| | Iteration | 3 |
| Decoder | Spatial Broadcast | Repeat slot $\rightarrow (width \times height)$ |
| | Positional Embedding | Added to broadcasted slots |
| | Conv $5 \times 5$ | 32 channels, stride 1, padding 2, ReLU |
| | Conv $5 \times 5$ | 32 channels, stride 1, padding 2, ReLU |
| | Conv $5 \times 5$ | 32 channels, stride 1, padding 2, ReLU |
| | Conv $3 \times 3$ | 4 channels (RGB + alpha), stride 1, padding 1 |

*Table 31.* Slot Attention architecture for CLEVR ($128\times128$) resolution.

| Module | Type / Operation | Details |
|---|---|---|
| Encoder | Conv $5 \times 5$ | 64 channels, stride 1, padding 2, ReLU |
| | Conv $5 \times 5$ | 64 channels, stride 1, padding 2, ReLU |
| | Conv $5 \times 5$ | 64 channels, stride 1, padding 2, ReLU |
| | Conv $5 \times 5$ | 64 channels, stride 1, padding 2, ReLU |
| | Positional Embedding | Added after conv block |
| | GroupNorm + Conv $1 \times 1$ | 64 channels, ReLU, Flatten |
| Slot Attention | Slot Initialization | Learnable $\mu, \sigma$ |
| | Slot | $D_{slots} = 64$ |
| | GRU Update | GRUCell (dim=64) per slot |
| | MLP Block | 2-layer FFN with ReLU, hidden dim = 128 |
| | Iteration | 3 |
| Decoder | Spatial Broadcast | Repeat slot $\rightarrow (8 \times 8)$ via repeat |
| | Positional Embedding | Added to broadcasted slot maps |
| | Transposed Conv $5 \times 5$ | 64 ch., stride 2, padding 2, output pad 1, ReLU |
| | Transposed Conv $5 \times 5$ | 64 ch., stride 2, padding 2, output pad 1, ReLU |
| | Transposed Conv $5 \times 5$ | 64 ch., stride 2, padding 2, output pad 1, ReLU |
| | Transposed Conv $5 \times 5$ | 64 ch., stride 2, padding 2, output pad 1, ReLU |
| | Conv $5 \times 5$ | 64 ch., stride 1, padding 2, ReLU |
| | Conv $3 \times 3$ | 4 ch. (RGB+mask), stride 1, padding 1 |

**Implementation of SlotMLP**   We follow SlotMLP from (Locatello et al., 2020). `SlotMLP` replaces Slot Attention module with Multi-Layer Perceptron (MLP). Table 32 demonstrated detailed implementations of SlotMLP.

*Table 32.* Architecture specification of the SlotMLP baseline.

| Module | Layer / Operation | Details |
|---|---|---|
| Encoder | Conv $5 \times 5$ | 32 channels, stride 1, padding 2, ReLU |
| | Conv $5 \times 5$ | 32 channels, stride 1, padding 2, ReLU |
| | Conv $5 \times 5$ | 32 channels, stride 1, padding 2, ReLU |
| | Conv $5 \times 5$ | 32 channels, stride 1, padding 2, ReLU |
| | Adaptive AvgPool2d | Resize to $(16 \times 16)$ |
| | Positional Embedding | Added to feature map before flatten |
| | GroupNorm + Conv $1 \times 1$ | Output shape: $(B, 32, 256)$ |
| SlotMLP | Input Flattening | Flatten to $(B, 32 \times 16 \times 16)$ |
| | 4-layer MLP | $512 \to 1024 \to 1024 \to K \times D$ |
| | Per-slot MLP | Applied independently to each slot |
| | Output | $(B, K, D)$ slot representations |
| Decoder | Spatial Broadcast | Repeat each slot $\to (D, H, W)$ |
| | Positional Embedding | Added to each slot map |
| | Conv $5 \times 5$ | 32 channels, stride 1, padding 2, ReLU |
| | Conv $5 \times 5$ | 32 channels, stride 1, padding 2, ReLU |
| | Conv $5 \times 5$ | 32 channels, stride 1, padding 2, ReLU |
| | Conv $3 \times 3$ | 4 ch. (RGB+mask), stride 1, padding 1 |

**Implementation of Slot Attention without GRU**   . The outputs of the attention module (referred as `updates` from (Locatello et al., 2020)) are updated using a Gated Recurrent Unit (GRU) (Chung et al., 2014) followed by a residual multi-layer perceptron (MLP) with normalization. We ablate the GRU from the Slot Attention. Outputs of the attention module (*updates*) are directly passed to residual MLP.

**Implementation of Slot Attention without Random Initialization**   Slot Attention produces a set of $K$ *slots* $\in \mathbb{R}^{K \times D_{slots}}$, where each slot is initially sampled from the Gaussian distribution with learnable $\mu$ and $\sigma$. To evaluate the effects of random initialization of slots, we trained a modified version of Slot Attention which consists of learnable embeddings. Instead of initializing $\in \mathbb{R}^{K \times D}$ from $\mathcal{N}(\mu, \sigma)$, we use $_{embed} \in \mathbb{R}^{K \times D_{slot}}$ as slot initial representations and follow the Slot Attention process. We adopt BO-QSA (Jia et al.) for this analysis, an advanced Slot Attention employing learnable queries and bi-level optimization.

**Implementation of Slot Attention without MLP**   The outputs of the attention module (referred as `updates` from (Locatello et al., 2020)) are updated using a Gated Recurrent Unit (GRU) (Chung et al., 2014) followed by a residual multi-layer perceptron (MLP) with normalization. We ablate the residual MLP from the Slot Attention. Outputs of the GRU module are directly referred as slots for each iteration (and outputs) and passed to next iteration without MLP.

**Variational Auto Encoder**   Implementation of Variational Auto Encoder (VAE) (Kingma et al., 2013) is demonstrated at Table 33. Following (Dittadi et al., 2021), we set the dimension of the latent vector to $D_{latent} = D_{slot} \times K$, where $K$ refers to the number of slots (objects including the background) introduced in the single image.

**Learning without forgetting**   Learning without forgetting (LwF) (Li & Hoiem, 2017) is a regularization-based continual learning method that preserves knowledge (knowledge distillation) of previous tasks by maintaining the output predictions (logits) of the old model through distillation, without storing any past data. It enables the model to learn new tasks while minimizing performance degradation on old tasks using a combination of cross-entropy and knowledge distillation losses. However, implementing LwF to Slot Attention (SA+LwF) requires few modifications. Since Slot Attention trained by mean squared error (MSE), it does not have a classifier. So, instead of maintaining output predictions (logits), SA+LwF performs knowledge distillation by maintaining the output of decoder $\mathcal{G}$.

$$\mathcal{L} = \lambda \cdot \mathcal{L}_{new}(X_n, \hat{X_n}) + \mathcal{L}_{old}(X_o, \hat{X_o}) \tag{7}$$
$$\text{where } \hat{X_n} = \mathcal{G}_o(\mathcal{SA}_s(\mathcal{F}_s(X_n)))$$
$$\text{and } \hat{X_n} = \mathcal{G}_n(\mathcal{SA}_s(\mathcal{F}_s(X_n)))$$

*Table 33.* Architecture specification of the Baseline VAE.

| Module | Type / Operation | Details |
|---|---|---|
| Encoder | Conv 1 | Conv2D(3 → 64,kernel=5,stride=2,pad=2),LeakyReLU |
| | ResBlock | Conv2D(64 → 64), BN, LeakyReLU |
| | ResBlock | Conv2D(64 → 64), BN, LeakyReLU |
| | Downsample | Conv2D(64 → 128, stride=2) |
| | ResBlock | Conv2D(128 → 128), BN, LeakyReLU |
| | ResBlock | Conv2D(128 → 128), BN, LeakyReLU |
| | Downsample | Conv2D(128 → 128, stride=2) |
| | ResBlock | Conv2D(128 → 256), BN, LeakyReLU |
| | ResBlock | Conv2D(256 → 256), BN, LeakyReLU |
| | Flatten | $256 \times H' \times W'$ to 1D |
| | MLP | Linear($\cdot \to 512$) + LayerNorm + LeakyReLU |
| Decoder | Input Layer | Conv2D(256 → 256,kernel=5,pad=2),LeakyReLU |
| | ResBlock | Conv2D(256 → 256), BN, LeakyReLU |
| | ResBlock | Conv2D(256 → 256), BN, LeakyReLU |
| | Upsample | ConvTranspose2D(256 → 128, stride=2) |
| | ResBlock | Conv2D(128 → 128), BN, LeakyReLU |
| | ResBlock | Conv2D(128 → 128), BN, LeakyReLU |
| | Upsample | ConvTranspose2D(128 → 64, stride=2) |
| | ResBlock | Conv2D(64 → 64), BN, LeakyReLU |
| | ResBlock | Conv2D(64 → 64), BN, LeakyReLU |
| | Final Conv | Conv2D(64 → 3, kernel=5, pad=2), Sigmoid |

We utilize parameters of decoder as task specific parameters and parameters of encoder $\mathcal{F}$ and Slot Attention $\mathcal{SA}$ as shared parameters. We train SA+LwF by Equation 8 with additional weight decay (5e-04). We follow details from (Li & Hoiem, 2017), and performed grid search to find the best $\lambda$ for SA+LwF 24 and use the value of $\lambda$=1, softmax temperature 2.0.

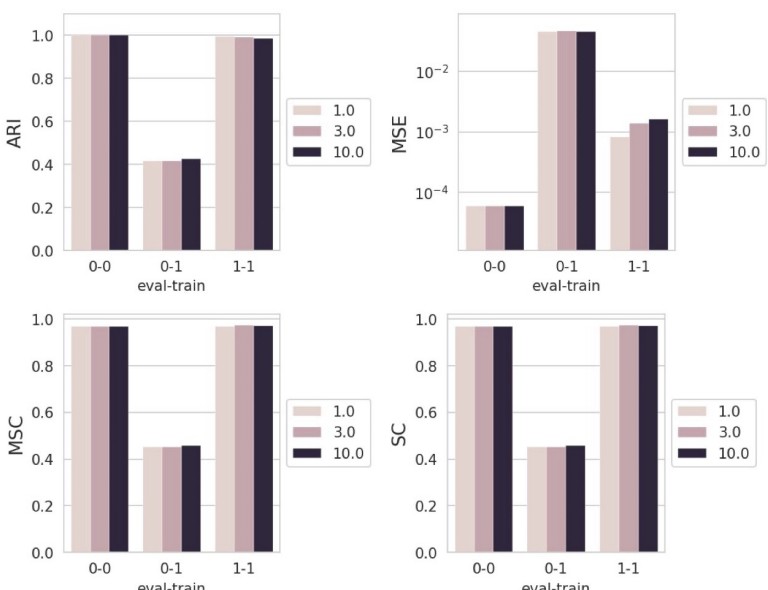

*Figure 24.* SA+LwF performance for different balancing parameter $\lambda$.

**Elastic Weight Consolidation** Elastic Weight Consolidation (EWC) (Kirkpatrick et al., 2017) is a regularization-based continual learning method that prevents forgetting by penalizing changes to important weights for previous tasks. To constrain important parameters to stay close to their old values, EWC implements this constraints as a quadratic penalty. It uses the Fisher Information Matrix to estimate the importance of each parameter. This helps the model retain old knowledge while learning new tasks.

$$\log p(\theta \mid \mathcal{D}) = \log p(\mathcal{D}_B \mid \theta) + \log p(\theta \mid \mathcal{D}_A) - \log p(\mathcal{D}_B) \tag{8}$$

In order to justify this choice of constraint and to define which weights are most important for a task, EWC consider neural network training from a probabilistic perspective. Equation 8 shows that the approximation of the posterior from task A as a prior for task B, helping the model preserve knowledge from previous tasks while learning new ones. However, since true posterior probability is intractable, EWC approximate the posterior as a Gaussian distribution with mean given by the parameters $\theta_A$ and a diagonal precision given by the diagonal of the Fisher information matrix $F$.

$$\mathcal{L} = \mathcal{L}_B(\theta) + \sum_i \frac{\lambda}{2} F_i(\theta_i - \theta_{A,i}^*)^2 \tag{9}$$

Equation 9 is objective of EWC. $\lambda$ is an hyper-parameter, which balances between old task and new one based on the importance of each task. We follow the implementation of EWC (Kirkpatrick et al., 2017) and train (SA+EWC). For $\lambda$, EWC use $\{0.7, 10, 25, 90\}$ for sequential MNIST, CIFAR10, and mini-ImageNet dataset. However, For Slot Attention which use MSE loss, we found that previous settings are not effective. We performed a grid search (Fig. 25), and use $\lambda=1e+04$ for SA+EWC.

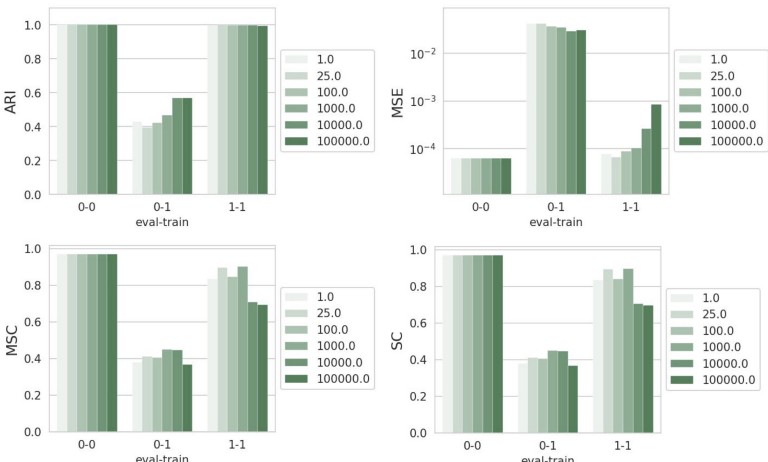

*Figure 25.* SA+EWC performance for different balancing parameter $\lambda$.

**Experience Replay** We apply standard Experience Replay (ER) to Slot Attention (SA+ER). Following prior work (Lopez-Paz & Ranzato, 2017a; Rebuffi et al., 2017; Chaudhry et al., 2019; Buzzega et al., 2020a), we perform *joint training* by concatenating replay samples with the current task's training data. SA+ER maintains a replay buffer that stores reconstruction results, similar to DPR (Post Replay), and uses the same buffer size. However, unlike DPR, which updates only the decoder after training, SA+ER performs joint training by sampling previous task data from the buffer and mixing it with current task data at the mini-batch level to mitigate catastrophic forgetting. Following (Lopez-Paz & Ranzato, 2017a), we concatenate replay and current samples in equal proportion. Details of training implementation of joint training are in Table 28.

**Generative Replay** We also evaluate a generative variant (SA+GR) inspired by Generative Replay (GR) methods (Shin et al., 2017; Wu et al., 2018; Ayub & Wagner; Zhai et al., 2019; Gao & Liu, 2023), which utilize auxiliary generative models (e.g., GANs (Shin et al., 2017), VAEs (Shin et al., 2017), or diffusion models (Gao & Liu, 2023)) to synthesize replay samples. In our implementation, we adopt a Variational Autoencoder (VAE) (Kingma et al., 2013) as the generative model, following (Shin et al., 2017), and use the architecture described in Table 33. Replay samples are generated from a Gaussian prior $\mathcal{N}(0, 1)$ and matched in quantity to those used in DPR. As in ER, we perform joint training by concatenating generated samples with the current task data in each mini-batch. Details of training implementation of joint training are in Table 28.

## B.5. Evaluation Metrics

**Adjusted Rand Index (ARI)** ARI (Rand, 1971) shows the segmentation quality of each slot via interpreting each segmentation mask as clustering. It quantifies the similarity between predicted segmentation of slots and ground-truth segmentation using clustering evaluation metrics. It corrects the Rand Index (RI) by accounting for chance agreement. The ARI ($\uparrow$) ranges from $-1$ to $1$, where 1 indicates perfect clustering, 0 indicates random labeling, and negative values indicate worse than random.

Let $a$ be the number of pairs of elements that are in the same cluster in both the predicted and ground-truth partitions, and $b$ be the number of pairs in different clusters in both. Then the ARI is defined as:

$$\text{ARI} = \frac{\sum_{ij} \binom{n_{ij}}{2} - \left[ \sum_i \binom{a_i}{2} \sum_j \binom{b_j}{2} \Big/ \binom{n}{2} \right]}{\frac{1}{2} \left[ \sum_i \binom{a_i}{2} + \sum_j \binom{b_j}{2} \right] - \left[ \sum_i \binom{a_i}{2} \sum_j \binom{b_j}{2} \Big/ \binom{n}{2} \right]} \tag{10}$$

where:

- $n_{ij}$ is the number of elements in both predicted cluster $i$ and ground-truth cluster $j$,

- $a_i = \sum_j n_{ij}$, the number of elements in predicted cluster $i$,

- $b_j = \sum_i n_{ij}$, the number of elements in ground-truth cluster $j$,

- $n$ is the total number of samples.

Following details from (Locatello et al., 2020) and only consider masks of foreground objects for computing ARI.

**Mean Squared Error (MSE)**  Mean Squared Error (MSE) quantifies the average squared difference between an original input image and its reconstructed version. We use the MSE score as an measure to evaluate the semantic accuracy captured by the model through its reconstruction quality

Given the input image $x \in \mathbb{R}^{H \times W \times C}$ and the reconstructed image $\hat{x} \in \mathbb{R}^{H \times W \times C}$, the MSE is defined as:

$$\text{MSE} = \frac{1}{HWC} \sum_{i=1}^{H} \sum_{j=1}^{W} \sum_{k=1}^{C} (x_{ijk} - \hat{x}_{ijk})^2 \tag{11}$$

where $H$, $W$, and $C$ denote the height, width, and number of channels of the image, respectively.

**Segmentation Covering (SC) and mean Segmentation Covering (mSC)**  Segmentation Covering (SC) (Arbelaez et al., 2010) evaluates how well each ground-truth segment is covered by the best-matching predicted segment. Higher SC($\uparrow$)

Given ground-truth segments $G = G_1, G_2, \ldots, G_m$ and predicted segments $P = P_1, P_2, \ldots, P_n$, the SC score is computed as:

$$\text{SC}(G, P) = \frac{1}{\sum_i |G_i|} \sum_i \max_j \frac{|G_i \cap P_j|}{|G_i \cup P_j|} \tag{12}$$

where:

- $|G_i|$ is the number of pixels in ground-truth segment $G_i$,

- $|G_i \cap P_j|$ is the intersection (overlap) between $G_i$ and predicted segment $P_j$,

- $|G_i \cup P_j|$ is the union of the two segments.

Mean Segmentation Covering (mSC) (Engelcke et al.) is an extended version of SC. mSC computes the average covering score across all images in a dataset. It quantifies how well predicted segments align with ground-truth segments over the entire evaluation set.

Given a dataset of $N$ images, where $G^{(i)}$ and $P^{(i)}$ denote the ground-truth and predicted segment sets for image $i$, the mSC is defined as:

$$\text{mSC} = \frac{1}{N} \sum_{i=1}^{N} \text{SC}(G^{(i)}, P^{(i)}), \quad \text{where } \text{SC}(G, P) = \frac{1}{\sum_k |G_k|} \sum_k \max_j \frac{|G_k \cap P_j|}{|G_k \cup P_j|} \tag{13}$$

where:

- $|G_k|$ is the number of pixels in the $k$-th ground-truth segment,

- $|G_k \cap P_j|$ is the overlap between ground-truth segment $G_k$ and predicted segment $P_j$,

- $|G_k \cup P_j|$ is their union.

A high SC (↑) score indicates good alignment and accurate localization of objects in the predicted segmentation. mSC (↑) values indicate better and robust segmentation quality and object alignment across images.

**Mean Best Overlap (mBO)**  The mean Best Overlap (mBO) (Pont-Tuset et al., 2016) measures the quality of object discovery by evaluating the overlap between predicted slot masks and ground-truth object masks. For each ground-truth object, we compute the Intersection-over-Union (IoU) with all predicted masks and take the maximum IoU as the best match. The mBO (↑) is then defined as the average of these best-match IoUs across all ground-truth objects in the dataset. A higher mBO indicates that predicted slots more accurately align with true objects, regardless of label permutation. Formally, let $\mathcal{G} = g_1, \ldots, g_{|\mathcal{G}|}$ denote the set of ground-truth masks and $\mathcal{P} = p_1, \ldots, p_{|\mathcal{P}|}$ the set of predicted masks. The best overlap score for a ground-truth object $g_i$ is defined as:

$$\text{BO}(g_i) = \max_{p_j \in \mathcal{P}} \frac{|g_i \cap p_j|}{|g_i \cup p_j|}, \tag{14}$$

where $|\cdot|$ denotes the number of pixels. The mean Best Overlap is then given by:

$$\text{mBO} = \frac{1}{|\mathcal{G}|} \sum_{i=1}^{|\mathcal{G}|} \text{BO}(g_i). \tag{15}$$

# C. Appendix: Continual Object-Centric Learning Benchmarks

In this section, we demonstrate our Continual Object-Centric benchmark. The goal is to evaluate the ability of object-centric methods on the task of unsupervised object discovery. we introduce two benchmarks: (1) Continual-Tetrominoes, and (2) Continual-CLEVR. These datasets build upon the original Tetrominoes and CLEVR (Johnson et al., 2017). Details on these datasets are discussed in the following sections.

Each dataset consists stream of tasks $\mathcal{T} = \{\mathcal{T}_t\}_{t=1}^{N}$, where $N$ is the the total number of tasks. Each $\mathcal{T}_t$ comprises multi-object images, with additional semantic labels for each object. During training unsupervised object discovery, the model is not given semantic labels. In C-OCL, novel object classes ($\mathcal{C}_t$) are incrementally introduced across tasks, where object classes of each task $\mathcal{T}_t$ are mutually exclusive across tasks (*disjoint*), ensuring that objects presented in task $\mathcal{T}_t$ never appear in any previous or future task. In this work, we focus on introducing novel *shape* classes, as shape provides a broader range of variation compared to *position* or *color*, which are limited to bounded continuous ranges.

## C.1. Training scenarios

We adopt three training and evaluation scenarios inspired by prior work (Shmelkov et al., 2017; Michieli & Zanuttigh, 2019; Cermelli et al., 2020), with modifications tailored for object-centric learning. In all scenarios, we ensure that at the initial task is consisted of five classes and at least two distinct object classes are introduced whenever novel classes are presented. The scenarios are defined as follows: (1) *Single Step addition of Two classes* (SST), (2) *Single Step addition of Multiple classes* (SSM), and (3) *Multi Step addition of Two classes* (MST) per step. Table 34 demonstrates details of each training scenarios.

*Table 34.* Task configuration for C-OCL under three scenarios.

| Setting | Task 0 | | | Task 1 | | | Task 2–5 | | |
|---------|--------|-------|------|--------|-------|------|-----------------|-------|------|
| | $\mathcal{C}_0$ | Train | Eval. | $\mathcal{C}_1$ | Train | Eval. | $\mathcal{C}_{2\sim5}$ | Train | Eval. |
| SST | 5 | 25000 | 5000 | 2 | 10000 | 5000 | – | – | – |
| SSM | 5 | 25000 | 5000 | 5 | 10000 | 5000 | – | – | – |
| MST | 5 | 25000 | 5000 | 2 | 10000 | 5000 | 2 | 10000 | 5000 |

## C.2. Continual-Tetrominoes

Our C-Tetrominoes dataset builds upon the original Tetrominoes (Kabra et al., 2019), which consists of images containing 3 Tetris pieces placed on a black background. The original dataset includes 19 different Tetris shapes and 6 distinct colors. Some of these shapes are symmetric under rotation or reflection.

In our work, we follow the 6 canonical Tetris shapes used in Tetrominoes (Kabra et al., 2019), excluding those with symmetric equivalence. To introduce additional object classes, we incorporate shapes from the Pentomino dataset (Montero et al.), which originally contains twelve pentomino shapes. From these, we select 9 distinct shapes to ensure visual and semantic diversity.

In total, our C-Tetrominoes dataset consists of 15 object shapes, each rendered in one of 10 different colors, placed on a uniform black background. Object placement is limited to 8 fixed positions per image to control for spatial variability.

We adopt the terminology from (Dittadi et al., 2021) to define semantic attributes in C-Tetrominoes. Specifically, the semantic labels consist of:

- *Shape*: 15 foreground object classes + 1 background class

    - 0 background class
    - 1 'Horizontal I'
    - 2 'Horizontal Z'
    - 3 'L pointing downward'
    - 4 'T pointing upward'
    - 5 'L pointing right'

- 6 'O'
- 7 'F'
- 8 'T'
- 9 'U'
- 10 'Z'
- 11 'X'
- 12 'L'
- 13 'N'
- 14 'P'
- 15 'V'

- *Color*: 10 foreground color classes + 1 background class

  - 0 $(0, 0, 0)$
  - 1 $(227, 74, 51)$
  - 2 $(227, 170, 55)$
  - 3 $(190, 222, 67)$
  - 4 $(101, 222, 101)$
  - 5 $(77, 213, 202)$
  - 6 $(81, 133, 244)$
  - 7 $(138, 91, 254)$
  - 8 $(203, 91, 232)$
  - 9 $(226, 70, 127)$
  - 10 $(228, 60, 105)$

- *Position*: 8 foreground spatial bins + 1 background class

Our C-OCL benchmarks have three different evaluation scenarios. We introduce details of each scenarios in the following paragraphs. For each dataset, we provide semantic labels of each object, with additional *mask* for each object.

**C-Tetrominoes Single Step addition of Two classes**   C-Tetrominoes Single Step addition of Two classes (SST) consist of two sequential tasks, introducing two new object shapes. Other factors except *shape* are identically shared across tasks.

*Table 35.* Configuration for C-Tetrominoes SST.

| Task | Shape | Color | Position | Object | Background | Train | Eval |
|------|-------|-------|----------|--------|------------|-------|------|
| $\mathcal{T}_0$ | 1, 2, 3, 4, 5 | 1–10 | x, y | 3 | 1 | 25,000 | 5,000 |
| $\mathcal{T}_1$ | 6, 7 | 1–10 | x, y | 3 | 1 | 10,000 | 5,000 |

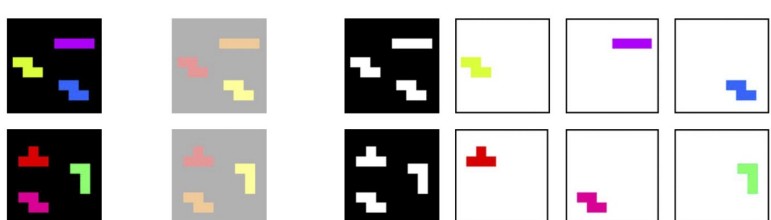

*Figure 26.* Examples of images of C-Tetrominoes SST $\mathcal{T}_0$. Starting from *left*, original image, concatenated mask, individual objects.

**C-Tetrominoes Single Step addition of Multiple classes**   C-Tetrominoes Single Step addition of Multiple classes (SSM) consist of two sequential tasks, introducing five new object shapes. Other factors except *shape* are identically shared across tasks.

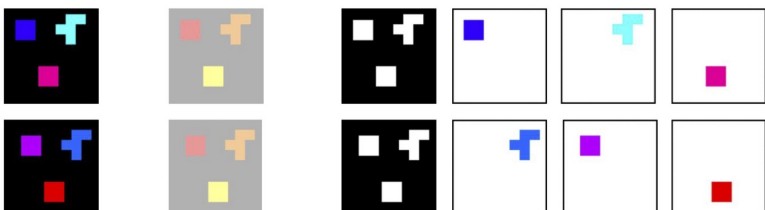

*Figure 27.* Examples of images of C-Tetrominoes SST $\mathcal{T}_1$. Starting from *left*, original image, concatenated mask, individual objects.

*Table 36.* Configuration for C-Tetrominoes SSM.

| Task | Shape | Color | Position | Object | Background | Train | Eval |
|------|-------|-------|----------|--------|------------|-------|------|
| $\mathcal{T}_0$ | 1, 2, 3, 4, 5 | 1–10 | x, y | 3 | 1 | 25,000 | 5,000 |
| $\mathcal{T}_1$ | 6, 7, 8, 9, 10 | 1–10 | x, y | 3 | 1 | 10,000 | 5,000 |

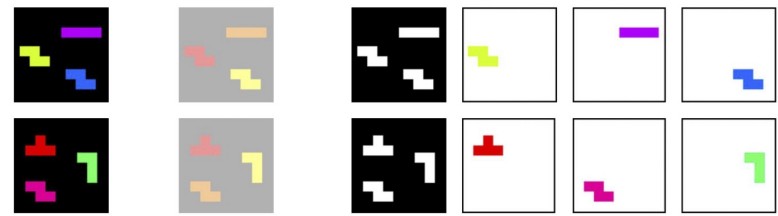

*Figure 28.* Examples of images of C-Tetrominoes SSM $\mathcal{T}_0$. Starting from *left*, original image, concatenated mask, individual objects.

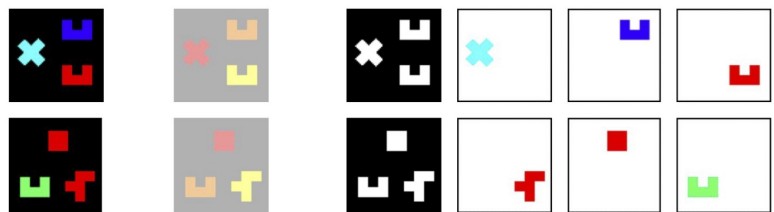

*Figure 29.* Examples of images of C-Tetrominoes SSM $\mathcal{T}_1$. Starting from *left*, original image, concatenated mask, individual objects.

**C-Tetrominoes Multi Step addition of Two classes** C-Tetrominoes Multi Step addition of Two classes (MST) consist of multiple sequential tasks, introducing two new object shapes. Other factors except *shape* are identically shared across tasks.

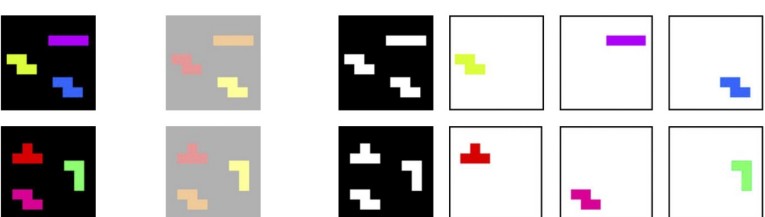

*Figure 30.* Examples of images of C-Tetrominoes MST $\mathcal{T}_0$. Starting from *left*, original image, concatenated mask, individual objects.

## C.3. Continual-CLEVR

Our C-CLEVR dataset builds upon the CLEVR dataset (Johnson et al., 2017), which features synthetic 3D scenes composed of up to 10 objects placed on a uniform gray background. Unlike Tetrominoes, CLEVR incorporates object occlusions, providing a more complex visual setting. The original CLEVR dataset includes 3 object shapes, 6 colors, continuous $(x, y)$ positions, 2 sizes, 2 materials, and object rotations. The CATER dataset (Girdhar & Ramanan, 2020), an extension of CLEVR designed for evaluating spatiotemporal reasoning, introduces two additional object shapes.

*Table 37.* Configuration for C-Tetrominoes MST.

| Task | Shape | Color | Position | Object | Background | Train | Eval |
|------|-------|-------|----------|--------|------------|-------|------|
| $\mathcal{T}_0$ | 1, 2, 3, 4, 5 | 1–10 | x, y | 3 | 1 | 25,000 | 5,000 |
| $\mathcal{T}_1$ | 6, 7 | 1–10 | x, y | 3 | 1 | 10,000 | 5,000 |
| $\mathcal{T}_2$ | 8, 9 | 1–10 | x, y | 3 | 1 | 10,000 | 5,000 |
| $\mathcal{T}_3$ | 10, 11 | 1–10 | x, y | 3 | 1 | 10,000 | 5,000 |
| $\mathcal{T}_4$ | 12, 13 | 1–10 | x, y | 3 | 1 | 10,000 | 5,000 |
| $\mathcal{T}_5$ | 14, 15 | 1–10 | x, y | 3 | 1 | 10,000 | 5,000 |

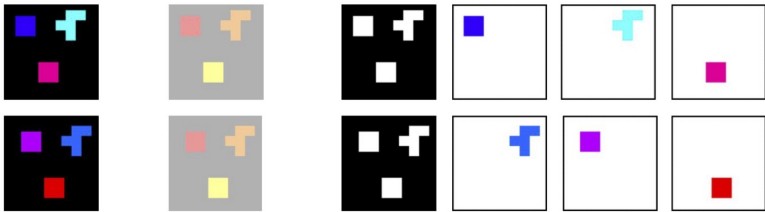

*Figure 31.* Examples of images of C-Tetrominoes MST $\mathcal{T}_1$. Starting from *left*, original image, concatenated mask, individual objects.

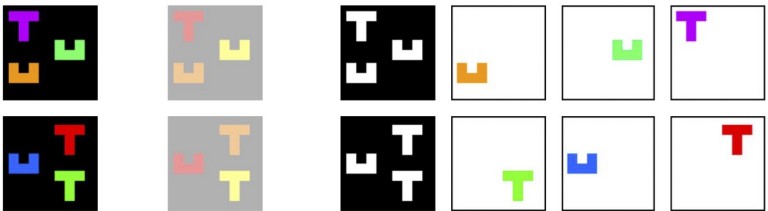

*Figure 32.* Examples of images of C-Tetrominoes MST $\mathcal{T}_2$. Starting from *left*, original image, concatenated mask, individual objects.

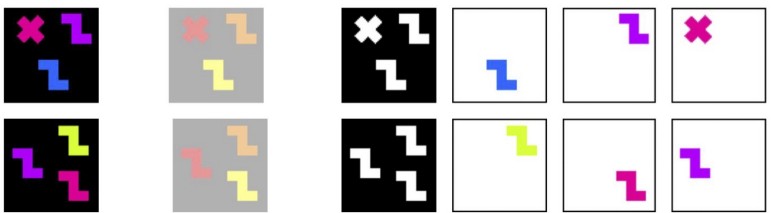

*Figure 33.* Examples of images of C-Tetrominoes MST $\mathcal{T}_3$. Starting from *left*, original image, concatenated mask, individual objects.

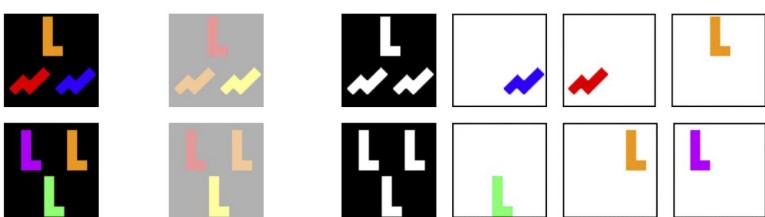

*Figure 34.* Examples of images of C-Tetrominoes MST $\mathcal{T}_4$. Starting from *left*, original image, concatenated mask, individual objects.

In constructing our C-CLEVR dataset, we incorporate 5 object shapes drawn from CLEVR and CATER, while preserving the original semantic attributes. To simulate continual learning, we introduce additional object classes that are incrementally added across tasks.

In total, C-CLEVR contains 15 object shapes, each rendered in one of 6 colors, 3 sizes, and 2 material types, with placement on a continuous $(x, y)$ space over a gray background. Similar to CLEVR6, a subset of CLEVR, our C-CLEVR always consists of 6 objects in the synthetic 3D scene.

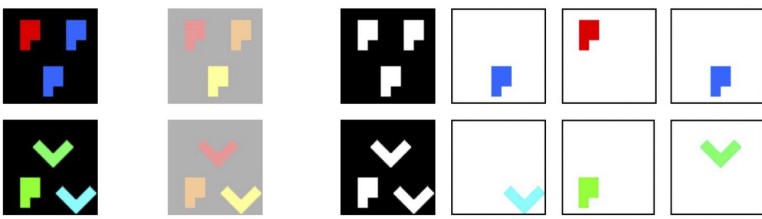

*Figure 35.* Examples of images of C-Tetrominoes MST $\mathcal{T}_5$. Starting from *left*, original image, concatenated mask, individual objects.

- *Shape*: 15 foreground object classes + 1 background class

    - 0 background class
    - 1 "cube"
    - 2 "sphere"
    - 3 "cylinder"
    - 4 "cone"
    - 5 "spl"
    - 6 "hourglass"
    - 7 "pudding"
    - 8 "tetrahedron"
    - 9 "octahedron"
    - 10 "dodecahedron"
    - 11 "icosahedron"
    - 12 "cross"
    - 13 "stellateddodecahedron"
    - 14 "torus"
    - 15 "spring"

- *Color*: 6 foreground color classes + 1 background class

    - 0 "gray" $(87, 87, 87)$
    - 1 "red" $(173, 35, 35)$
    - 2 "blue" $(42, 75, 215)$
    - 3 "green" $(29, 105, 20)$
    - 4 "brown" $(129, 74, 25)$
    - 5 "purple" $(129, 38, 192)$
    - 6 "cyan" $(41, 208, 208)$
    - 7 "yellow" $(255, 238, 51)$

- *Position*

    - x
    - y

- *Size*

    - "large"
    - "medium"
    - "small"

- *Materials*

    - "metal"

– "rubber"

Our C-OCL benchmarks have three different evaluation scenarios. We introduce details of each scenarios in the following paragraphs. For each dataset, we provide semantic labels of each object, with additional *mask* for each object.

**C-CLEVR Single Step addition of Two classes**    C-CLEVR Single Step addition of Multiple classes (SSM) consist of two sequential tasks, introducing two new object shapes. Other factors except *shape* are identically shared across tasks.

*Table 38.* Configuration for C-CLEVR SST.

| Task | Shape | Color | Position | Object | Background | Train | Eval |
|---|---|---|---|---|---|---|---|
| $\mathcal{T}_0$ | 1, 2, 3, 4, 5 | 1–7 | x, y | 6 | 1 | 25,000 | 5,000 |
| $\mathcal{T}_1$ | 6, 7 | 1–7 | x, y | 6 | 1 | 10,000 | 5,000 |

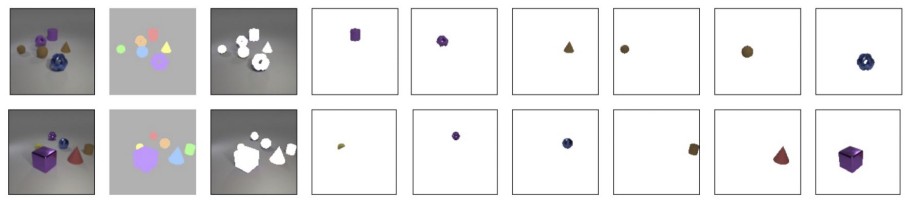

*Figure 36.* Examples of images of C-CLEVR SST $\mathcal{T}_0$. Starting from *left*, original image, concatenated mask, individual objects.

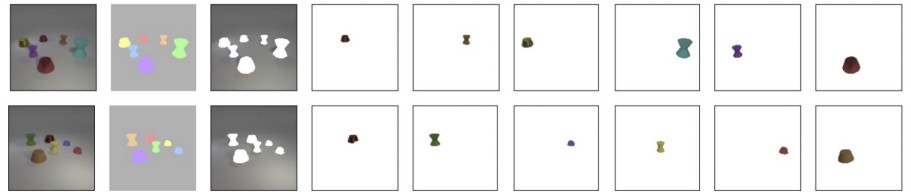

*Figure 37.* Examples of images of C-CLEVR SST $\mathcal{T}_1$. Starting from *left*, original image, concatenated mask, individual objects.

**C-CLEVR Single Step addition of Multiple classes**    C-CLEVR Single Step addition of Multiple classes (SSM) consist of two sequential tasks, introducing five new object shapes. Other factors except *shape* are identically shared across tasks.

*Table 39.* Configuration for C-CLEVR SSM.

| Task | Shape | Color | Position | Object | Background | Train | Eval |
|---|---|---|---|---|---|---|---|
| $\mathcal{T}_0$ | 1, 2, 3, 4, 5 | 1–10 | x, y | 6 | 1 | 25,000 | 5,000 |
| $\mathcal{T}_1$ | 6, 7, 8, 9, 10 | 1–11 | x, y | 6 | 1 | 10,000 | 5,000 |

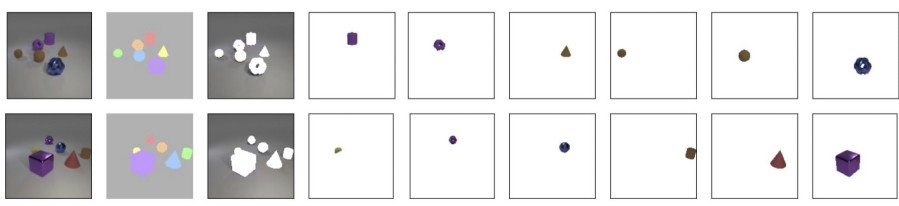

*Figure 38.* Examples of images of C-CLEVR SSM $\mathcal{T}_0$. Starting from *left*, original image, concatenated mask, individual objects.

**C-CLEVR Multi Step addition of Two classes**    C-CLEVR Multi Step addition of Two classes (MST) consist of multiple sequential tasks, introducing two new object shapes. Other factors except *shape* are identically shared across tasks.

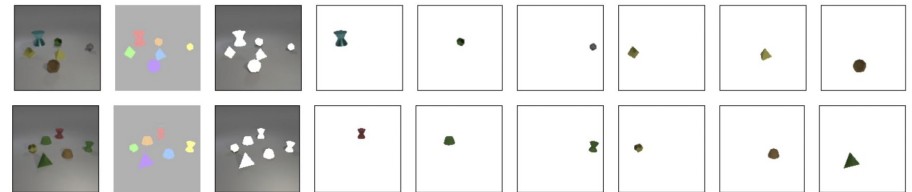

*Figure 39.* Examples of images of C-CLEVR SSM $\mathcal{T}_1$. Starting from *left*, original image, concatenated mask, individual objects.

*Table 40.* Configuration for C-CLEVR MST.

| Task | Shape | Color | Position | Object | Background | Train | Eval |
|------|-------|-------|----------|--------|-----------|-------|------|
| $\mathcal{T}_0$ | 1, 2, 3, 4, 5 | 1–10 | x, y | 6 | 1 | 25,000 | 5,000 |
| $\mathcal{T}_1$ | 6, 7 | 1–10 | x, y | 6 | 1 | 10,000 | 5,000 |
| $\mathcal{T}_2$ | 8, 9 | 1–10 | x, y | 6 | 1 | 10,000 | 5,000 |
| $\mathcal{T}_3$ | 10, 11 | 1–10 | x, y | 6 | 1 | 10,000 | 5,000 |
| $\mathcal{T}_4$ | 12, 13 | 1–10 | x, y | 6 | 1 | 10,000 | 5,000 |
| $\mathcal{T}_5$ | 14, 15 | 1–10 | x, y | 6 | 1 | 10,000 | 5,000 |

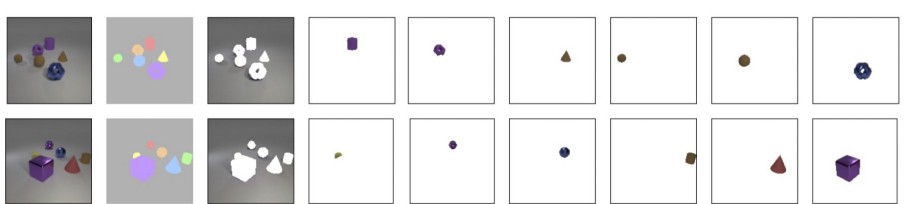

*Figure 40.* Examples of images of C-CLEVR MST $\mathcal{T}_0$. Starting from *left*, original image, concatenated mask, individual objects.

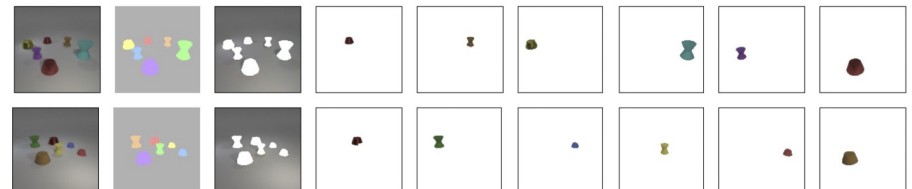

*Figure 41.* Examples of images of C-CLEVR MST $\mathcal{T}_1$. Starting from *left*, original image, concatenated mask, individual objects.

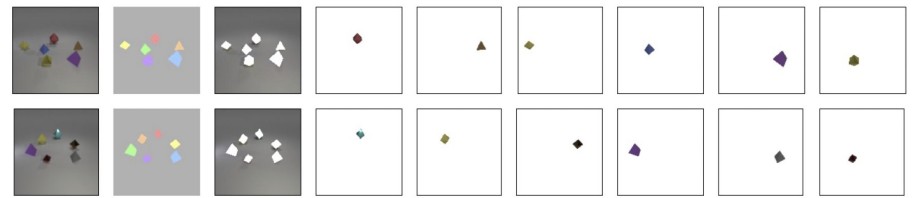

*Figure 42.* Examples of images of C-CLEVR MST $\mathcal{T}_2$. Starting from *left*, original image, concatenated mask, individual objects.

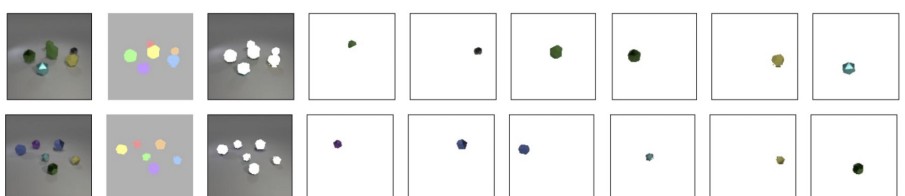

*Figure 43.* Examples of images of C-CLEVR MST $\mathcal{T}_3$. Starting from *left*, original image, concatenated mask, individual objects.

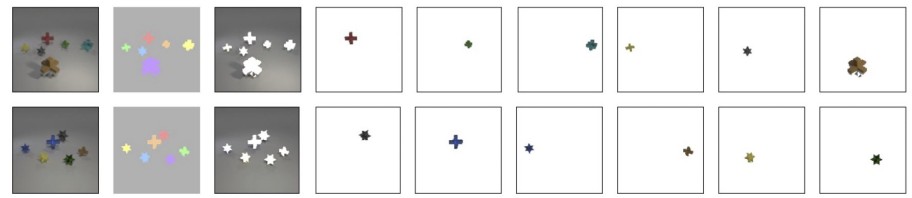

*Figure 44.* Examples of images of C-CLEVR MST $\mathcal{T}_4$. Starting from *left*, original image, concatenated mask, individual objects.

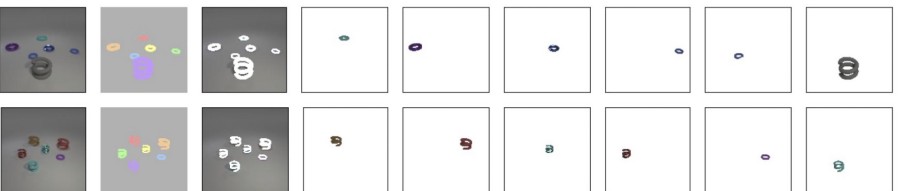

*Figure 45.* Examples of images of C-CLEVR MST $\mathcal{T}_5$. Starting from *left*, original image, concatenated mask, individual objects.

